

# Introducing PebbleCounts: A grain-sizing tool for photo surveys of dynamic gravel-bed rivers

Benjamin Purinton[1] and Bodo Bookhagen[1]

[1]Institute of Earth and Environmental Science, Universität Potsdam, Potsdam, Germany

**Correspondence:** Benjamin Purinton (purinton@uni-potsdam.de)

**Abstract.** Grain-size distributions are a key geomorphic metric of gravel-bed rivers. Traditional measurement methods include manual counting or photo sieving, but these are achievable only at the 1–10 $m^2$ scale. With the advent of unmanned aerial vehicles and increasingly high-resolution cameras, we can now generate orthoimagery over hectares at sub-cm resolution. These scales, along with the complexity of high-mountain rivers, necessitate different approaches for photo sieving. As opposed to other image segmentation methods that use a watershed approach to automatically segment entire images, our open-source algorithm, *PebbleCounts*, relies on k-means clustering in the spatial and spectral domain and rapid manual selection of well-delineated grains. The result is improved grain-size estimates for complex river-bed imagery, without any post processing. In a second step, we develop a fully automated method, *PebbleCountsAuto*, that relies on edge detection and filtering suspect grains, without the k-means clustering or manual selection steps. The algorithms are tested in controlled indoor conditions on three arrays of pebbles and then applied to $12 \times 1$ $m^2$ orthomosaic clips of high-energy mountain rivers collected with a camera-on-mast setup (akin to a low-flying drone). A 20-pixel b-axis length lower truncation is necessary for attaining accurate grain-size distributions. For the k-means *PebbleCounts* approach, average percentile bias and precision are 0.03 and 0.09 $\psi$, respectively, for $\sim$1.16 mm/pixel images, and 0.07 and 0.05 $\psi$ for one 0.32 mm/pixel image. The automatic approach has higher bias and precision of 0.13 and 0.15 $\psi$, respectively, for $\sim$1.16 mm/pixel images, but similar values of $-0.06$ and 0.05 $\psi$ for one 0.32 mm/pixel image. For the automatic approach, only at best 70% of the grains are correct identifications, and typically around 50%. *PebbleCounts* operates most effectively at the 1 $m^2$ scale, where the algorithm can be rapidly applied in $\sim$5 minutes in many small areas to acquire accurate grain-size data over 10–100 $m^2$ areas. These data can be used to validate *PebbleCountsAuto* applied at the scale of entire survey sites ($10^2$–$10^4$ $m^2$). We synthesize results and recommend best practices for image collection, orthomosaic generation, and grain-size measurement using both algorithms.

## 1 Introduction

Gravel-bed rivers transport water, nutrients, and sediment downstream, linking high mountains to populated forelands. The grain-size distributions (GSDs) — and associated percentile diameters, such as the $D_{50}$ and $D_{84}$ — in a river reach are fundamental geomorphic metrics of these systems (e.g., Shields, 1936; Parker et al., 1982; Church et al., 1998). They are used to characterize aquatic habitats (e.g., Kondolf and Wolman, 1993), assess the impacts of human infrastructure like dams (e.g., Kondolf, 1997; Grant, 2012), calibrate theoretical models of river transport and erosion (e.g., Sklar et al., 2006; Attal and Lavé,



2006; Attal et al., 2015; Dunne and Jerolmack, 2018), and explore natural phenomena such as downstream fining (e.g., Paola et al., 1992; Ferguson et al., 1996; Rice and Church, 1998; Gomez et al., 2001; Chatanantavet et al., 2010; Lamb and Venditti, 2016), which is essential for nutrient transport and ecological diversity.

Accurate grain-size measurement is elusive in nature given the heterogeneity of gravel-bed rivers, particularly in steep mountain catchments where the range of grain sizes is large. Traditionally, GSDs have been gathered via physical clast measurement and counting along grids (Wolman, 1954), lines (Wohl et al., 1996), or in ∼1 m$^2$ patches (Bunte and Abt, 2001), all truncated at some lower observable limit (e.g., Rice and Church, 1998). Not only are these techniques time consuming, prone to operator bias, and disruptive to the environment, but they also require large (hundreds of pebbles) sample sizes to accurately estimate the characteristic nature of the grains in each location (Wolcott and Church, 1991).

In light of this, measurement from photographs is an attractive option for increasing sample size and decreasing fieldwork, while covering larger areas. The advent of unmanned aerial vehicles (UAVs), or drones, and orthorectified photo-mosaic generation using Structure from Motion with Multi-View Stereo (SfM-MVS) (Smith et al., 2015), combined with increasingly affordable high-resolution — 12–24 megapixel (MP) — cameras, allows the collection of high-quality photo surveys at scales of entire river cross sections or reaches at resolutions at or exceeding 1 cm/pixel (Woodget and Austrums, 2017; Woodget et al., 2018). Even higher resolution (1 mm/pixel) river surveys over areas of $10^2$–$10^4$ m$^2$ can be accomplished with low flying UAVs, pole-mounted cameras, or using handheld imagery, and many of the steps associated with data collection and processing can be at least partially automated.

We build on previous work and introduce the addition of color-space clustering techniques to present efficient new semi-automated (*PebbleCounts*) and fully automated (*PebbleCountsAuto*) algorithms for grain identification and sizing in high-energy mountain rivers. Our algorithms are built on Python with a few popular libraries and are open source. The instructions and code can be accessed at: https://github.com/UP-RS-ESP/PebbleCounts (Purinton and Bookhagen, 2019). In this study, we present previous work on grain-size measurement from rivers and our motivation for new developments. The processing chains of *PebbleCounts* and *PebbleCountsAuto* are then discussed. We test the algorithms in controlled conditions and then in a more challenging field setting in the northwestern Argentine Andes. The limits and caveats of the method are discussed using imagery of varying resolution, and suggestions for photo collection and processing are provided.

## 2 Previous Work on Photo Sieving

Manual digitization of each pebble was previously necessary for grain sizing from pictures (e.g., Kellerhals and Bray, 1971; Ibbeken and Schleyer, 1986). Modern digital grain sizing is divided into texture- and segmentation-based image-processing methods. Texture methods rely on the relationship between grains and their shadowed interstices to derive size estimates over image windows. Examples include semivariance (Verdú et al., 2005; Carbonneau et al., 2003, 2004; Carbonneau, 2005), entropy or inertia calculated from gray level co-occurrence matrices (GLCM) (Haralick et al., 1973; Carbonneau et al., 2004; Carbonneau, 2005; Dugdale et al., 2010; de Haas et al., 2014; Woodget and Austrums, 2017; Woodget et al., 2018), and



autocorrelation (Rubin, 2004; Warrick et al., 2009; Buscombe et al., 2010). These methods only provide one estimate of grain size (e.g., $D_{50}$), which often requires site-specific calibration.

Buscombe (2013) achieved full GSD measurements using wavelet decomposition on gray-scaled sand and pebble imagery, and also published their technique as an open-source Python tool. This is another texture method that does not measure each

grain individually, and it is more apt for thin sections or beach sands, since it requires that each grain be fully resolvable and that the distributions be relatively homogeneous in size and shape. An additional texture method relies on the 3D texture (or roughness) of point clouds to relate the variance of bed-scale topography to average grain size (Rychkov et al., 2012; Westoby et al., 2015; Woodget and Austrums, 2017; Bertin and Friedrich, 2016), however, this technique also requires site calibration and the relationships have been found to vary widely depending on, among other things, grain sorting and packing (Pearson

et al., 2017).

In contrast to texture methods, the focus of segmentation is the full delineation and measurement of every visible grain. Segmentation is error prone in images that contain overlapping grains, a large range of grain sizes including sand patches, changes in landcover (e.g., vegetation), pebbles that are highly irregular in shape (non-ellipsoid), pebbles with intra-granular color variations or texture such as veins or fractures, and in which shadowing is irregular. Herein, we refer to these factors col-

lectively as image complexity. The benefits are that segmentation does not require any site calibration besides knowledge of the image scale and it provides a full GSD and all the commonly used percentiles ($D_{5,16,25,50,75,84,95}$). Published methods include the work of Butler et al. (2001), Sime and Ferguson (2003), and Graham et al. (2005a, b), all of which rely on edge detection followed by watershed segmentation and ellipse fitting to each separate grain region to get the long (a) and intermediate (b) grain axes. Detert and Weitbrecht (2012) added some sophistication to the edge detection and watershed steps of Graham et al.

(2005a, b) and provide a free — though closed source — application called *Basegrain* for the commercial software package *Matlab$^{TM}$*, which has become a standard tool (e.g., Bertin and Friedrich, 2016; Bertin et al., 2017; Langhammer et al., 2017; Carbonneau et al., 2018).

## 3   Motivation for New Methods

Watershed segmentation is effective for interlocking, uniformly colored, oblate grains, however, energetic gravel-bed rivers in

mountains often have more complex grain compositions with intra-granular variation, irregular shadowing, and a large range of sizes. The automated watershed methods proposed suffer from over-segmentation, grain misidentification, and the need for significant post-processing when applied to complex images.

In the interest of attaining GSDs from these settings and in images with a mix of clasts and sand patches, we are motivated to develop a new semi-automated technique that uses k-means clustering of pixels and rapid manual selection of well-defined

grains, herein referred to as the K-means with Manual Selection (KMS) or *PebbleCounts* approach, and a fully automated version that uses filtering of suspect grains, herein referred to as the Automatic with Image Filtering (AIF) or *PebbleCountsAuto* approach (Fig. 1). By avoiding over-segmentation and misidentification associated with the watershed approach, we are able to



select fewer grains per image, but be sure that those selected are correctly delineated, thus improving the resulting GSD (Fig. 2), with the intention of up-scaling to include many thousand grain measurements over large areas.

Furthermore, faced with diverse camera models and the rise of SfM-MVS for the generation of georeferenced orthophotos, we wish to explore reasonable and appropriate combinations for covering hectare-sized areas while maintaining accurate 5 measurement of characteristic GSDs. Fundamentally, our aim for the KMS approach is not in the delineation of a single high-resolution image from a $\sim 1$ m$^2$ patch as in previous segmentation work, but rather a method that can cover areas of 10–100 m$^2$ containing complex grain arrangements, despite missing many grains at the patch scale. These semi-automated photo-sieving results can then be used to validate the AIF method at much greater spatial scales ($10^2$–$10^4$ m$^2$). This work serves as both a presentation of a new algorithm and a guide for the successful collection of GSDs in complex mountainous settings over large 10 survey areas, where physical grain sizing is not feasible and previously reported image processing methods are unreliable or time consuming.

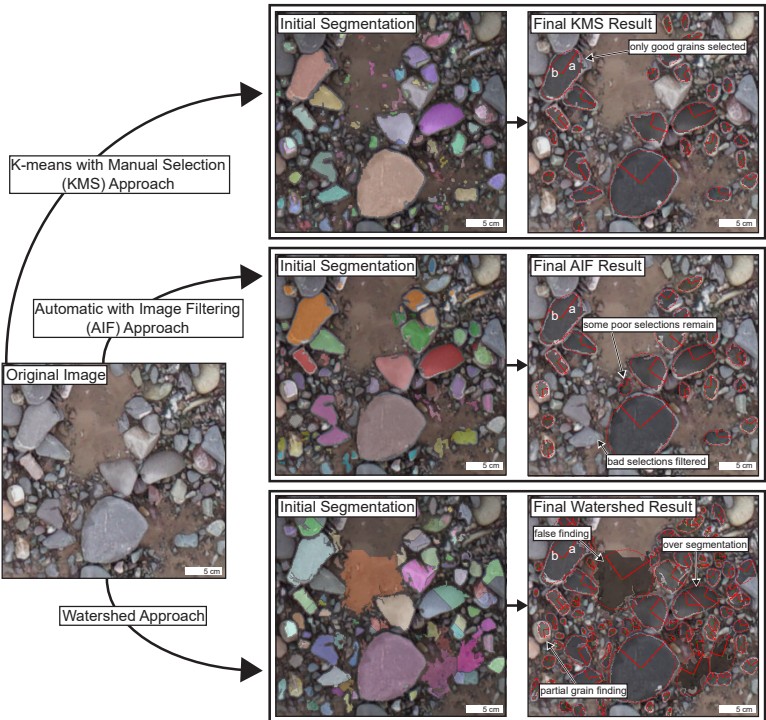

**Figure 1.** The conceptual difference between our K-means with Manual Selection (KMS) and Automatic with Image Filtering (AIF) approaches versus a fully automated watershed segmentation approach on a gravel image from a high-mountain river. The a- and b-axes of each grain mask are found via an ellipse fit to the same area. Fewer grains are found in the KMS and AIF results, and there is still some misidentification in the case of AIF, but less than in the watershed result.





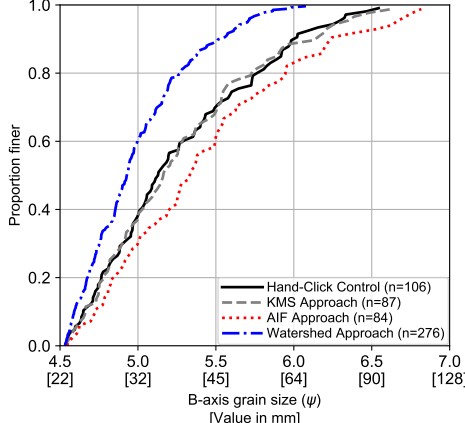

**Figure 2.** Watershed segmentation (blue, dashed and dotted line) versus KMS (gray, dashed line) and AIF (red, dotted line) approaches compared with a hand-clicked b-axis GSD (black line) for a ~1 m$^2$ river patch (S09 in Figure 9b). Watershed approach leads to over-segmentation of grains, giving an unreasonable number of clasts (276 versus 106 in the control) and an overly fine GSD.

## 4 The Algorithms

The methods developed here hold similarities to previous work by Graham et al. (2005a) and Detert and Weitbrecht (2012), with some key differences. Processing is presented briefly, and we direct the interested user to the manual for a full description of the steps: https://github.com/UP-RS-ESP/PebbleCounts (Purinton and Bookhagen, 2019).

### 4.1 *PebbleCounts*: K-means with Manual Selection (KMS)

The general outline of *PebbleCounts* is shown in Figure 3. We employ the additional color spaces HSV (hue, saturation, value) and CIELab (Russ, 2002), aside from traditional RGB (red, green, blue) and gray-scale, to enhance differences in the spectral domain separate from lighting. First, the RGB image undergoes strong non-local means denoising (Buades et al., 2011) to smooth intra-granular color difference, interactive gray-scale shadow masking (Otsu, 1979) to separate obvious interstices, and HSV color selection for sand-patch masking. The image and shadow/sand mask are then windowed for further processing. At each window, the RGB image undergoes another weaker non-local means denoising, is then converted to CIELab, and the chromaticity bands from this color space undergo bilateral filtering (Tomasi and Manduchi, 1998) to preserve inter-granular edges while further smoothing color. Following this, edge detection on the smoothed, gray-scaled image occurs via a combination of top-hat, Sobel, and Canny methods with feature-AND selections (Russ, 2002), in which an edge is added to the full mask only if it overlaps with a found edge in the shadow-, sand-, or previous edge-mask, thus piece-wise building an edge map while avoiding lone (i.e., intra-granular) edges (Detert and Weitbrecht, 2012).

After edge detection, our algorithm uses k-means clustering (Lloyd, 1982; Sculley, 2010) to further segment the pebbles. First, the matrix of non-masked pixels is converted into a vector that includes the spectral information at each location. This





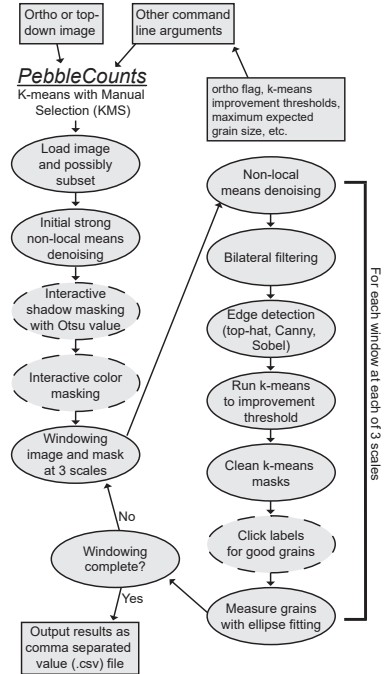

**Figure 3.** Flowchart of *PebbleCounts*. The boxes are user supplied input or output from the algorithm. Dashed lines indicate a user input step during processing, either entering and checking values or clicking.

$N \times 4$ dimensional vector ($N$ being the number of non-masked pixels) includes two spectral observables: the green-red and blue-yellow smoothed chromaticity bands from CIELab; and the two spatial observables: the $x$ and $y$ coordinates of the pixel in image space. To avoid over-segmentation by anisotropic or image-spanning grains, the $x, y$ coordinates are rescaled to 50% of the color, which is also rescaled from 0 to 1. We attempted using agglomerative Ward hierarchical clustering (Ward, 1963)

to further improve results on anisotropic and/or large grains, however, this approach is prohibitively slow on large images, and test results did not show significant improvement. K-means clustering requires a user-supplied number of clusters. Here, we add clusters beginning at 1 and recalculate the k-means clustering up to an inertia improvement threshold of 1–10%. The resulting k-means labeled masks are cleaned via binary operations and the user is prompted to select the labeled regions that contain full, single grains within a simple pop-up window.

After selection, the orientation and a- and b-axes of an ellipse fit to the labeled region, shown to accurately approximate grain size (Graham et al., 2005a), are recorded and the grain is added to the final list and the masked region. This processing takes place over three separate scales representing a "burrowing" of the algorithm through the image (from largest to smallest window/grain size). Scales are set by the user supplied longest expected a-axis and image resolution. In contrast to the 46 variables employed by *Basegrain*, *PebbleCounts* has 20 command-line variable flags — of which 15 exert influence on the

results — with most requiring little to no modification (Table S1). Examples of the command-line interface and manual clicking steps are shown in Figure 4 and Figure 5, respectively.



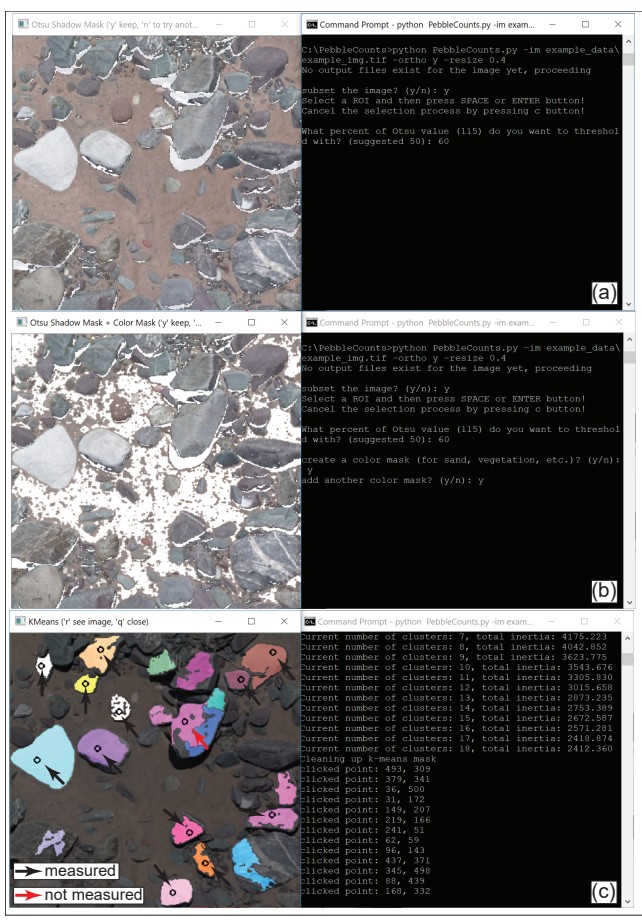

**Figure 4.** Example of command-line and pop-up interface for *PebbleCounts*. (a) Interactive Otsu thresholding using percentage of Otsu value and yes ('y') or no ('n') confirmation. (b) Interactive color masking by yes ('y') or no ('n') and resulting color mask after selection. (c) K-means clustering and pop-up window for pebble selection by left clicking, with black arrows measured in final output and red arrows ignored after right-click removal (see Fig. 5).



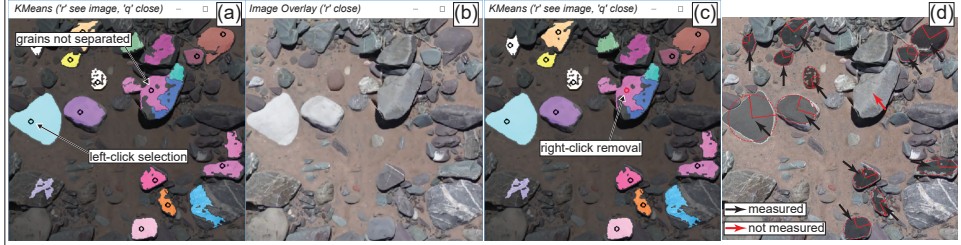

**Figure 5.** Clicking tutorial continued from Figure 4c. Following k-means clustering at each scale a mask overlaid on the original image is presented (a), and grains are selected by a left click anywhere in the segmented area, resulting in a black circle at the click location. When clicking is finished the mask is closed by pressing 'q'. To view the original unmasked image the user may press 'r' (b). Using this switching the user can see which grains are poorly delineated and remove the last click with a right click on the mouse (c). The original black circle selection turns to red to signify this grain is off and will not be measured in the final output (d).

### 4.2 *PebbleCountsAuto*: Automatic with Image Filtering (AIF)

The general outline of *PebbleCountsAuto* is shown in Figure 6. This method applies the same initial non-local means denoising and interactive shadow/sand masking, with the option to input user supplied values for full automation. From here, we diverge from the windowing and k-means approach and move directly to edge detection on the entire image using the same top-hat,

Canny, and Sobel combination with feature-AND selections.

The resulting mask is then cleaned via binary morphological operations (e.g., erosion and dilation) and each disconnected label in the resulting mask is measured as a grain via ellipse fitting. To reduce the misidentified grains, the ellipses are filtered in a three-step chain: (A) Does the centroid fall within another ellipse?; (B) Does the ellipse overlap with any neighboring ellipses above some threshold?; and (C) Is the percent misfit (ellipse area vs. grain-mask area) above some threshold? At

each step, an answer of yes leads to the elimination of the grain. The (A) and (B) steps filter grains that have high overlap or are over-segmented, whereas (C) helps filter areas where multiple grains were combined in one mask or a non-grain was identified (e.g., remaining sand patch). Only the remaining, unfiltered grains are taken as the final results, with the assumption of higher uncertainties, but that the remaining misidentified grains are minimal compared to the good grains, particularly when up-scaling to large areas and tens-of-thousands of pebbles on high-quality (low-blur) images. The command-line variables for

this method are shown in Table S2, and the first steps are identical to Figure 4a,b.

We experimented with resampling (over- and under-sampling) the image prior to grain detection to increase smoothing and to improve the detection of larger grains at the cost of measuring fewer smaller grains. The majority of images achieved the best results using the original resolution, though we did find a slight improvement in results using under-sampling on some unsharp images (see Section S3 in the supplement). The selection of other parameters like the maximum percent misfit is also

covered in Section S3 in the supplement.



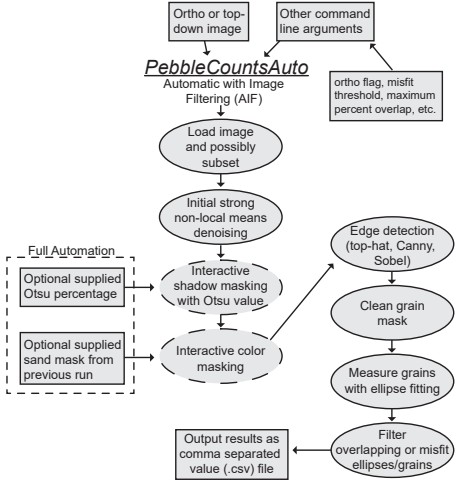

**Figure 6.** Flowchart of *PebbleCountsAuto*. The boxes are user supplied input or output from the algorithm. Dashed lines indicate a user input step during processing, either entering and checking values or clicking.

## 5   Calibration and Validation Test I: Controlled Experiment

### 5.1   Experimental Setup

To test the KMS and AIF approaches on a simple control we arranged three distributions of well-rounded, river pebbles with a-axis sizes from 3–130 mm in semi-overlapping patterns in a 0.5×0.5 m area (Fig. 7). As opposed to most studies that use b-axis lengths to measure the GSD (Bunte and Abt, 2001), in the experimental setup we use a-axes since it was easier to hand-measure the longest axis of each of the > 200 grains measured. Six size class bins (3–5, 10–20, 25–35, 40–50, 60–70, and 80–130 mm; all a-axis) were sampled to approximate two log-normal and one bimodal GSD. These classes ensured the clear demarcation of sizes into the appropriate binned values, irrespective of small uncertainties in measurement. The river pebbles were selected to have uniform intra-granular color with minimal striations (i.e., veins), low angularity, and a diverse array of inter-granular colors. Lighting was controlled by overhead fluorescent bulbs and the photos were taken without flash to limit cast shadows. The choice of background was a textured carpet surface to provide enough match points around the pebbles in SfM-MVS processing.

### 5.2   Camera Setup

We tested a Fujifilm X100F model camera with a fixed 23 mm focal length lens and a Sony α6000 model with a removable 35 mm fixed length lens. Both had the same advanced photo system type-C (APS-C) sensors (23.6 mm×15.6 mm) and both output photos at 24 MP in a 4000×6000-pixel format. Following initial tests, it became clear that the image quality and grain-size results were practically identical for these two cameras, so the results presented are only those for the Fujifilm, as the photo quality was slightly sharper throughout and less distorted at the image corners. To simulate reduced quality, the 24 MP



Fujifilm picture dimensions were reduced to 75, 50, and 25%, resulting in 13.5, 6, and 1.5 MP images at pixel dimensions of 3000×4500, 2000×3000, and 1000×1500, respectively.

## 5.3 Images

### 5.3.1 Top-down Images

To measure objects on images, the image scale (or resolution) must be known and effectively uniform throughout the area of interest. The simplest way to calculate top-down photo resolution is by the camera parameters and camera height, with the resolution in scene height ($H_r$) and width ($W_r$) in mm/pixel given by:

$$H_r = \frac{(S_H \cdot h)}{(f \cdot I_H)} \tag{1}$$

$$W_r = \frac{(S_W \cdot h)}{(f \cdot I_W)} \tag{2}$$

where $S_{H,W}$ is the sensor height and width in mm, $f$ is the lens focal length in mm, $h$ is the camera height in mm, and $I_{H,W}$ is the image height and width in pixels. This equation assumes no major distortions within the field of view, which is not valid

for oblique imagery, but is negligible for top-down photography at close range using non-fisheye lenses. With $h$=1.55 m, the resulting image resolutions tested from the Fujifilm were approximated at 0.26, 0.35, 0.53, and 1.05 mm/pix, with less than 0.01 mm/pixel difference in $H_r$ and $W_r$. Recalculation of resolution with variable camera height between 1.4 and 1.7 m ($\pm$ 0.15 m uncertainties) led to < 0.03 mm/pixel differences in resolution. Furthermore, these values were within 0.001 mm of the resolution of resulting orthomosaics from SfM-MVS processing of multiple overlapping images with input scale bars. Given

the negligible effect of distortion and differences in $H_r$ and $W_r$, we suggest the following simplifying equation for calculating top-down photo resolution ($R$):

$$R = \frac{H_r + W_r}{2} \tag{3}$$

### 5.3.2 Orthomosaic Images: SfM-MVS Processing

To ensure uniform resolution, we used multiple overlapping photos taken from different angles (up to 16 photos per setup, including at least 4 overhead shots) to generate SfM-MVS orthoimages in *Agisoft Photoscan v.1.4.2* (Agisoft, 2018) — re-

named *Agisoft Metashape* in recent versions. This allows rapid output of additional information including point clouds, digital elevation models (DEMs), and the undistorted orthomosaics, with resolution recorded in the image metadata for direct input into *PebbleCounts* and *PebbleCountsAuto*. *Agisoft* processing was carried out in the following steps:



1. Image quality detection and the exclusion of photos with quality metric < 0.7. This step analyzes pixel contrast to estimate sharpness with values ranging from 0 (blurred) to 1 (sharp). We found 0.7 to be a sufficient lower cutoff upon visual inspection of results.

2. Detection of 12-bit coded targets in the remaining photos, with two targets placed at each of the four corners of the area and ensuring that the diameter of the printed targets' center circle was limited to 10–30 pixels in image resolution for successful automated detection.

3. Input of scale for the orthomosaic output, provided by the distances between the targets at each corner (resulting in four distance measurements) with 0.5 mm accuracy using a ruler with cm and mm demarcations.

4. Photo alignment at high quality with a 40,000 key-point and 2000 tie-point limit.

5. Dense cloud generation from the aligned photos at the medium output and with moderate depth filtering. Given the high quality of the photos more aggressive options did not improve results.

6. DEM building from the dense cloud with default settings in a local coordinate system.

7. Generation of an orthomosaic from the input imagery and DEM at the default settings.

8. Output of the orthomosaic to a GeoTiff file with resolution provided in m/pixel.

## 5.4 Comparison Metrics

For the simple, controlled experiment, with relatively coarse grain-size bins, it is not appropriate to compare percentiles (e.g., $D_{50}$) or to run Kolmogrov-Smirnov (KS) tests and measure the difference in distributions between the AIF or KMS and control GSDs. Instead, we compared the counts in each bin between the control and algorithm and visually assessed the matching of the GSDs. This provides a reasonable baseline for checking the performance of the algorithm in a highly controlled setting.

## 5.5 Controlled Experiment Results

For each of the three 150–200 clast arrangements, the KMS *PebbleCounts* run time was ~7 minutes on a laptop with 16 GB RAM and 2 cores (Intel i7-6650U 2.20 GHz) and no GPU, whereas the AIF *PebbleCountsAuto* run time was ~1 minute. Both the top-down and orthoimagery was used, but the results were entirely consistent aside from some inter-run variability in the KMS approach caused by the non-unique solution of k-means clustering. Given this consistency, we only present the results from the top-down images. Furthermore, the use of only 4 top-down photos also generated the same results, albeit in about $1/6^{th}$ the processing time of using all 12–16 photos (~10 minutes versus ~1 hour on the same laptop).

Across all three distributions, the KMS approach consistently undercounts the number of clasts in each a-axis bin (Fig. 7). However, and in agreement with previous research (Graham et al., 2010), this undercounting is uniformly distributed and thus the GSDs do not show notable differences between the algorithm and control. For the two arrangements with increased fine (3–5 mm) and coarse (60–130 mm) pebbles (Fig. 7b,c), the undercounting is stronger at the finer end of the distribution leading



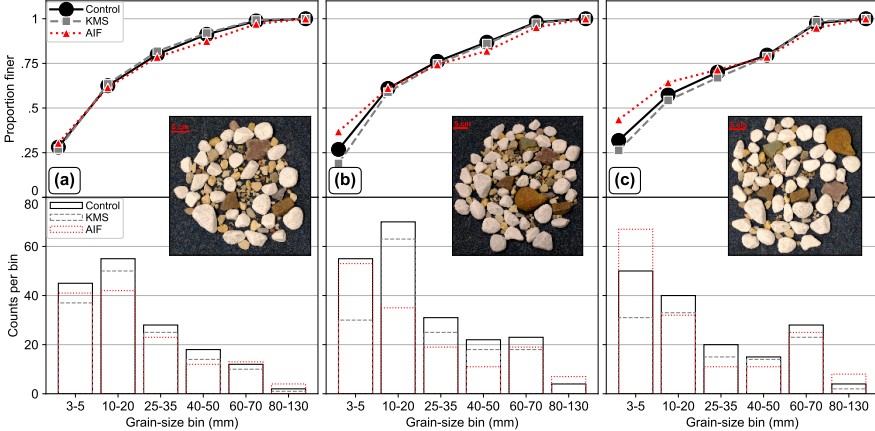

**Figure 7.** Result of KMS (gray, dashed lines) and AIF (red, dotted lines) on the three experimental lab setups (a-c) with known grain inputs in six size classes (black line), measured as the grain a-axis. (a) Log-normal, (b) log-normal with increased number of all classes, including fines, and (c) skewed bimodal with increased number of coarser grains. Bottom row shows the counts per bin and the top row shows the resulting GSD. The images are 0.26 mm/pixel (24 MP).

to a slight underestimation of the GSD by the KMS approach in this region. This is caused partially by the user missing more of the smaller grains (of which there are exponentially more), some smaller grains being partially hidden by the larger, and also by the smallest grains being only a few pixels in area and thus eliminated during mask-cleaning steps, or not captured at all. On the other hand, the AIF approach tends to overcount the fine pebbles, leading to overestimation of the GSD, because many small

non-grain areas remaining in the masked image are automatically selected in the final result, rather than ignored as in the KMS approach. As we reduced the resolution from 0.26–1.05 mm/pixel, the reduction in the finest size class increased dramatically for the KMS approach (Fig. 8). At the lowest resolution tested (1.5 MP), this undercounting leads to severe discrepancies in the GSD curve. As the resolution degrades it becomes more difficult to discern rocks in the smallest size class (3–5 mm), which correspond to an a-axis grain size of 12–19, 9–14, 6–9, and 3–5 pixels for the 24, 13.5, 6, and 1.5 MP resolution, respectively,

indicating the necessity of a limiting lower measurement factor (e.g., Graham et al., 2005a).

## 6 Calibration and Validation Test II: Field Surveys

### 6.1 Field Setting

Having established the algorithms on control data, we sought to evaluate the performance on complex, natural photos. Field data provides the real-world application and detailed uncertainty analysis most useful for researchers seeking to apply the

15 methods to their own sites. For this we turned to photo surveys carried out on gravel-bed river cross sections of the foreland and topographic transition zone of the northwestern Argentine Andes (Fig. 9). This is an area of strong precipitation, topographic, and environmental gradients, and the rivers surveyed are dynamic environments capable of transporting enormous quantities of



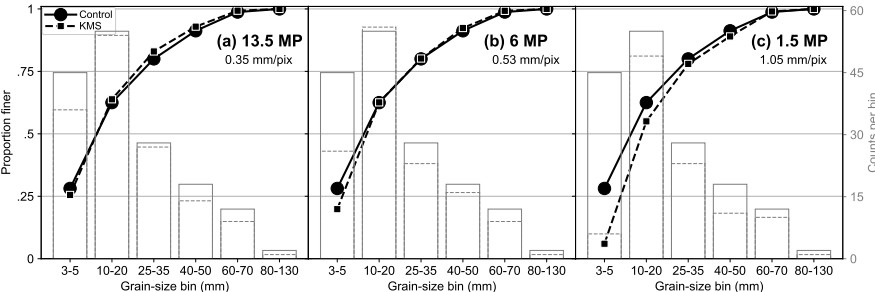

**Figure 8.** Results of reducing the image dimensions to (a) 75% (13.5 MP), (b) 50% (6 MP), and (c) 25% (1.5 MP) and re-running the KMS approach on the distribution in Figure 7a. Control is shown as black (left y-axis) and gray (right y-axis) solid lines and KMS as the dashed lines.

sand, gravel, and boulders of various lithology (Bookhagen and Strecker, 2012; Purinton and Bookhagen, 2018). Catchment-average erosion rates from the area, based on cosmogenic nuclide inventories, suggest rates on the order of 0.6–1 mm/yr (Bookhagen and Strecker, 2012), with large variability during the Pleistocene and Holocene (Tofelde et al., 2017). The region is frequently affected by extreme hydrometeorologic events that lead to flooding and drainage-pattern re-arrangement (Castino

et al., 2016, 2017).

## 6.2   Surveying and Orthomosaic Generation

All cross-section surveys were collected using the Sony $\alpha$6000 camera model at 24 MP, and survey sizes ranged from ~1000–5000 m$^2$. In this case, the standard zoom lens delivered with the camera was used at the shortest focal length of 16 mm to maximize the field of view. Also, to help cover the large survey sites, the camera was affixed to the end of a pole with a

remote control trigger, allowing overhead shots to be collected from a height of 4.5–5 m (Fig. 10), giving a ground resolution of approximately 1.1–1.2 mm/pixel by eq. (3). UAV flights have proven difficult in the windy conditions experienced in these valleys, but flights at 20–30 m heights with the 12 MP camera provided on the DJI Mavic and Phantom models (focal lengths of 3.6–4.3 mm, sensor dimensions of 6.17×4.55 mm, and image dimensions of 4000×3000 pixels) would result in image resolutions of ~7–13 mm/pixel, and are thus inadequate for delineating cm-scale pebbles.

To generate georeferenced orthomosaics that could be tiled and passed directly to *PebbleCounts* and *PebbleCountsAuto*, survey sites on the dry river-bed were laid out with on average 18 coded targets (with a range of 10–24) and the position of each was measured with a differential GPS (Fig. 10). Kinematic post-processing with a permanent base station < 100 km away at the Universidad Nacional de Salta (UNSA) in Salta, Argentina, led to cm accuracy of XYZ target locations. The site was traversed in a cross-hatched pattern with a photo captured every 2–3 paces, so that each location appeared in ~9 top-

down pictures from different angles. *Agisoft* processing is similar to that described for the experiment (see Section 5.3.2.), with some key differences. Here, the scale was provided by the XYZ coded target locations in UTM zone 19S, WGS84 ellipsoidal datum. Given the increased complexity of the setting and imperfect photo collection, the dense point cloud was





**Figure 9.** (a) Field cross-section survey sites (black triangles) in NW Argentina from three gravel-bed rivers (Toro, Vaqueros, and Grande) and their tributaries, draining from the sparsely vegetated mountains in the west towards the verdant foreland and city centers of Salta and Jujuy in the east. The Landsat 8 RGB composite satellite image (using bands 2, 3, and 4) from 12 June 2017 shows the climatic transition from wet foreland to dry mountains, demarcated by the green-brown transition zone corresponding to vegetation changes running approximately north-south. (b) Detailed view of the $12 \times \sim 1$ m$^2$ orthomosaic clips from each of the field sites with average resolution of 1.16 mm/pixel.



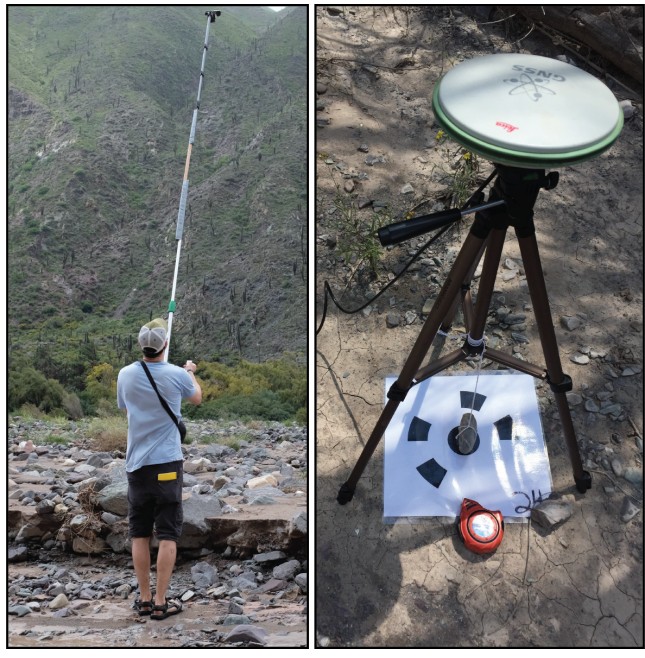

**Figure 10.** Sony $\alpha$6000 24 MP camera affixed to mast for photo collection at a height of 4.5–5 m (left) and differential GPS measurement of coded targets (right).

generated at high quality with aggressive depth filtering. The DEMs and orthomosaics were also output in UTM zone 19S projections, providing undistorted pixels with resolution in m/pixel. Given the volume of photos (600–1300 per site), the sites were processed automatically using the Python API for *Agisoft*, with processing times consistently over 10 hours on an 80 core, 500 GB RAM server making use of 1 GPU NVIDIA Tesla K80 unit for some of the steps (e.g., dense matching).

From 10 of our full survey sites over three different river systems we selected $12 \times \sim 1$ m$^2$ patches to clip out of the full orthomosaics and evaluate using the KMS and AIF approaches. The final resolution of these 12 GeoTiff orthoimages matched the theoretical value from eq. (3), with an average of 1.16 mm/pixel and range of 1.08–1.24 mm/pixel (standard deviation of 0.05 mm/pixel). The patches (Fig. 9b) include variable amounts of sand and a large range of grain sizes, packing arrangements, and shadowing. From one site (S14A) there were hand-held images available for the same selected patch from the same

Sony $\alpha$6000 camera zoomed to 20 mm focal length and taken from a height of $\sim$1.5 m, allowing for the generation of a complementary orthomosaic at 0.32 mm/pixel resolution.

## 6.3   Control Data and Comparison Metrics

For control data from the field we return to b-axis measurements (rather than a-axes as in the lab). In each patch, the b-axes of all grains visible to the naked eye were manually digitized. This generated a 5490 pebble control dataset across all 12 mast-

surveyed sites. For the lone hand-held patch at 0.32 mm/pixel, the control data was 1726 pebbles versus 621 from the same





patch at the 1.12 mm/pixel mast resolution, as smaller grains could be manually measured on the image at a 4-times improved resolution.

The use of continuous control data, as opposed to discrete bins in the lab experiment, allows a more detailed investigation of the performance of both approaches, including biases and their correction. B-axis measurements of overlapping control and

KMS grains were compared to look for sizing bias. This was followed by a search for the lower truncation limit (the lower cutoff in b-axis length in pixels that grains are reliably measured at) of the algorithm, also using the KMS results. For parts of the analysis, the size data were converted to the typical $\psi$ scale ($\psi = -\phi = log2(mm)$) of grain-size measurement of coarse river sediments. This allows direct comparison of statistical results with other studies (e.g., Graham et al., 2005b)

We compared the GSDs from the KMS and AIF approaches with the control using a two sample KS-test to check the null

hypothesis that the two samples are drawn from the same distribution. Because sample sizes were at times small, leading to erroneous KS-test results, we also devised a second metric of GSD comparison. Similar to the KS-test, which uses the maximum distance between the cumulative distribution functions (CDFs), or in our case the GSDs, our metric interpolates both distributions to the same lengths in 0.1 $\psi$ steps and then sums the difference between the re-interpolated curve to give an approximate integral of the difference between the two GSDs (AIF or KMS minus the control), which we term $A_{diff}$. Here,

an $A_{diff}$ value close to 0 indicates good matching, and positive or negative values indicate underestimation or overestimation, respectively.

We also examined the performance of some key percentiles ($D_{5,16,25,50,75,84,95}$). The metrics for comparison of control ($P_C$) and KMS or AIF ($P_P$) percentiles are consistent with other studies (Sime and Ferguson, 2003; Graham et al., 2005b, 2010). These are the mean ($m = \frac{1}{n} \cdot \Sigma(P_P - P_C)$), the mean squared ($ms = \frac{1}{n} \cdot \Sigma(P_P - P_C)^2$), and the irreducible random error

($e = \sqrt{ms - m^2}$). The bias of *PebbleCounts* is quantified by $m$, and $e$ measures the scatter or precision after bias correction (Sime and Ferguson, 2003).

## 6.4   Field Survey Results

### 6.4.1   Initial Results: Biases and Their Correction

The KMS *PebbleCounts* approach took ∼10 minutes per 1 m$^2$ orthomosaic clip at 1.16 mm/pixel resolution, depending on

the number of grains, and particularly the number of finer grains, present. Run time for the AIF *PebbleCountsAuto* approach was typically ∼2 minutes per site. All run times refer to the same laptop with 16 GB RAM and 2 cores (Intel i7-6650U 2.20 GHz) and no GPU. For the 0.32 mm/pixel image the processing for KMS took ∼45 minutes, as there were more fine grains to be identified (given the log-normal distribution) and so the clicking took exponentially longer, and the AIF took ∼20 minutes given the longer time spent filtering the large number of grains. These run times refer to the use of no lower truncation value

and only some morphological (e.g., erosion and dilation) cleaning operations. We note that the use of a GPU for the filtering steps will significantly improve processing time.

An aggregation and coarse binning of all b-axes in the control versus KMS and AIF data for the coarser imagery are presented in Figure 11. There is obvious undercounting in these data from the KMS results, similar to the experimental setup,





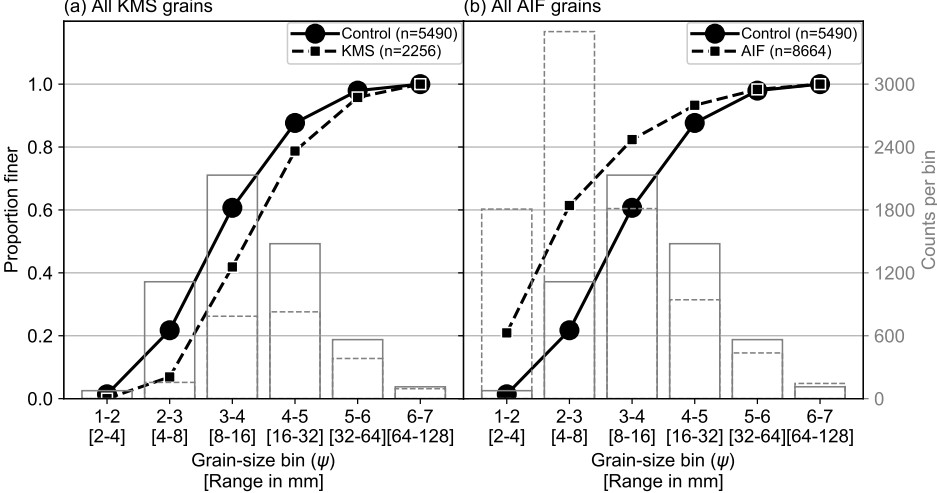

**Figure 11.** Comparison of (a) KMS and (b) AIF at the 12 field sites all aggregated and coarsely binned. Control is shown as black (left y-axis) and gray (right y-axis) solid lines and KMS and AIF as the dashed lines.

and it appears in this case to be causing a significant discrepancy in the GSD curves. Whereas the manual clicking found over 1000 grains in the smallest classes (1–2 and 2–3 $\psi$), the KMS approach found none in the smallest and only ∼100 in the second smallest. This skews the percentiles to the higher grain sizes, and thus overestimates them significantly. In opposition to this, but again in agreement with the experimental setup, the AIF results display significant overcounting at the finer sizes as many

non-grains are identified, particularly when the algorithm is run with no lower truncation.

The skewed results from both the KMS and AIF approaches warrant detailed analysis of the algorithms' deficiencies and GSD corrections. To begin, we examined the performance of *PebbleCounts* on grains manually digitized and the same grains selected during clicking in the KMS approach on the coarser imagery (Fig. 12). There is only a slight negative bias across all grain sizes, indicating underestimation of individual grains by *PebbleCounts*, however, this median shift varies with no apparent

pattern and is likely caused by uncertainties in the manual b-axis digitization of thousands of grains. For instance, digitization with b-axis vector lines can achieve sub-pixel accuracy compared to the raster processing of *PebbleCounts*. The AIF approach measures grains identically to the KMS method and thus has the same misfit errors on correctly identified grains. From this we conclude that the algorithm is effective on a grain-by-grain basis and the skewing of the GSDs are instead caused by sampling errors related to the image resolution and ability to find small grains (see Figure 8).

The undercounting error can be explored on the full distribution of pebbles by gradually increasing the lower truncation value and assessing the error in percentiles versus the control data at each step (Fig. 13). As truncation is increased, the median percentile error decreases rapidly up to an inflecting value — manually chosen from the graph as a significant local minimum — where the median difference is near 0 mm. Truncating the KMS distributions at a minimum b-axis length of 23 mm (rounded to 20 pixels) improves the results significantly for the 1.16 mm/pixel imagery taken from the mast. Beyond this truncation,



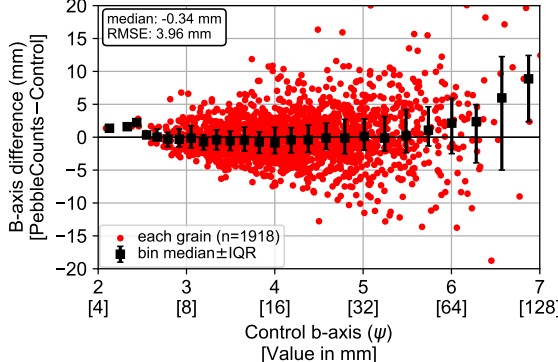

**Figure 12.** Measurement error of *PebbleCounts* (here the KMS results) versus control on a grain-by-grain basis for overlapping grains in the coarser (1.16 mm/pixel) imagery. There is an overall median shift, but the binned medians do not display a consistent pattern.

there is limited improvement. Regarding the 0.32 mm/pixel image, the 20-pixel (6.5 mm) truncation also results in a median difference near 0 mm, with subsequent truncation values leading to only ~0.5 mm improvements. Supplying these truncation values directly to the KMS *PebbleCounts* tool results in reduced processing time to ~5 minutes for the coarser imagery and ~15 minutes for the finer, as many small grains were then ignored and left out of the clicking mask.

The same analysis for the AIF approach is complicated by the large number of false grains found and the extreme over-counting of fine grains. Given this, we instead make the assumption that the similarity of the two methods, particularly in the edge detection and ellipse fitting steps, leads to similar errors in both. Therefore, we assume the same 20-pixel truncation. For the AIF *PebbleCountsAuto* tool, processing times with the 20-pixel truncation reduced to < 1 minute and ~3 minutes for the coarse and fine images, respectively.

**6.4.2    Results: Mast Images**

The combined results before and after lower truncation for the coarser (~1.16 mm/pixel) imagery taken from the mast surveys is shown in Figure 14. For separate plots of the 12 different sites before and after truncation in the KMS approach see Section S2 in the supplement. Without any lower truncation, the AIF tool results in significant overcounting and GSD underestimation with a high $A_{diff} > 8$. The KMS tool instead shows undercounting and GSD overestimation with a low $A_{diff} < -4$. Both have

KS-test $p$-values < 0.0001. When we apply a 20-pixel truncation, both the AIF and KMS approaches achieve $A_{diff}$ values near or below $-1$, with the manual KMS approach performing best and achieving a high KS-test $p$-value of 0.2398. The AIF approach retains a low $p$ of 0.0008 with a ~0.1–0.2 $\psi$ bias towards coarser values in the upper portion of the GSD (> $D_{50}$).

    In Figure 15, we show the 20-pixel truncated KMS and AIF results on a site-by-site basis. For the KMS approach, following truncation 11 sites have $p$-values > 0.1 and one site (S16) has $p$=0.0971. $A_{diff}$ values are also near 0 indicating close matching

of the GSDs, aside from S24 and S34, which both show large discrepancies. The AIF results in Figure 15 follow a similar trend





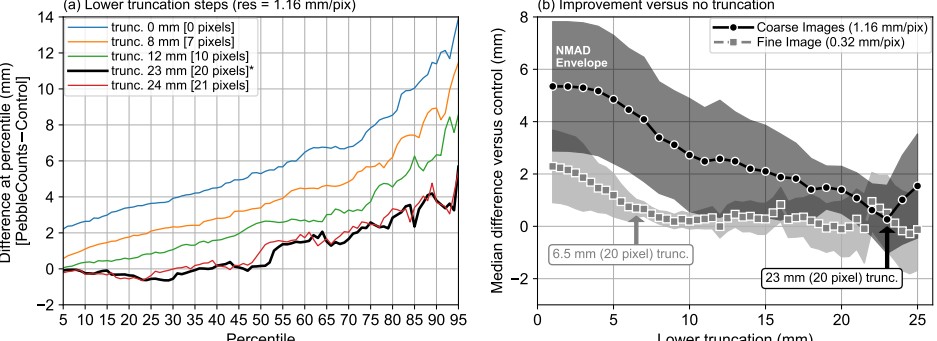

**Figure 13.** (a) Error in each percentile (5–95) as lower truncation value is increased in 1 mm steps for the 1.16 mm/pixel imagery. Only a few steps are plotted for clarity. (b) The median difference in percentiles compared with the control versus the lower truncation value, with the normalized median absolute difference (NMAD) shown as the error envelope (Höhle and Höhle, 2009). From this analysis, we select a lower truncation of 20 pixels. The analysis in (a) was repeated for the finer image (with 0.5 mm truncation steps) to get the gray squares line in (b), and is not shown here.

to the KMS results. The main difference is that, for the AIF approach, there is a bias towards coarser values, with many $A_{diff}$ values $< -1$, and generally poorer results compared with the KMS approach, with GSDs being overestimated by $\sim$0.1–0.2 $\psi$.

In the KMS results, despite a high $p$-value, S24 demonstrates a stronger bias in the GSD towards coarser grains (up to 0.5 $\psi$ discrepancy), as indicated by the high $A_{diff}$ value of $-1.36$. Here, the KS-test pass is likely caused by the small sample size remaining after truncation ($n$=24), the least of any site. The poor performance of S24 was expected given the large size range with many sub-cm pebbles and a few large boulders, strong cast shadows from the large grains, and intra-granular edges on angular boulders with quartz veins (see Figure 9b). Importantly, S24 is the only site not from a major river stem, but rather from a debris-flow fan draining a small tributary catchment in the Quebrada del Toro. S34 also had a high $A_{diff}$=$-2.11$. In this case, poor performance is due to significant blurriness of this image, and again a small sample size ($n$=47).

We also compared the individual percentiles of interest to assess the bias and accuracy of truncated results (Fig. 16). For the KMS approach, the bias ($m$) is 0.06 $\psi$ with a precision ($e$) of 0.13 $\psi$. Excluding S24 and S34, $m$ and $e$ drop to 0.03 and 0.09 $\psi$, respectively. The AIF results have higher $m$ and $e$ values of 0.15 and 0.17 $\psi$, respectively, which are reduced to 0.13 and 0.15 $\psi$ following exclusion of the same S24 and S34 sites, in addition to the S10 site, which was also somewhat blurry and with relatively few grains. For the AIF percentiles, we chose to include S16 despite large overestimation at higher percentiles (Fig. 15), as this was a sharp image with a relatively large sample size. The high uncertainties from this scene likely require some adjustment of the edge-detection variables (see Section S3 in the supplement) for improved segmentation, but the results presented are realistic for fast processing using the AIF method, with the caveat of higher expected uncertainties.

The uncertainties in Figure 16 are average values, and the inset plots also demonstrate the increasing uncertainty of larger percentiles. The maximum uncertainty for both at $D_{95}$ is $m$=0.08 $\psi$ and $e$=0.07 $\psi$ for the KMS result and $m$=0.35 $\psi$ and $e$=0.2 $\psi$ for the AIF result. Importantly, since the $\psi$ scale is logarithmic, the larger errors at higher percentiles correspond to similar





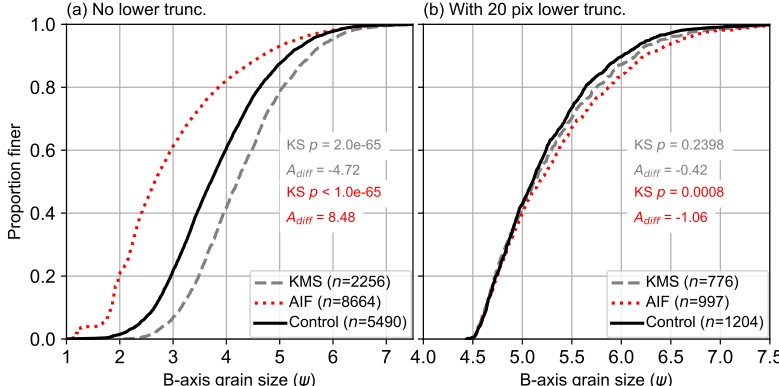

**Figure 14.** Results from hand-clicked control (black line), KMS *PebbleCounts* (gray, dashed line), and AIF *PebbleCountsAuto* (red, dotted line) with the initial non-truncated run (a) and the 20-pixel truncated run (b). In corresponding colors are the $p$-value results of a KS-test and the $A_{diff}$ approximate integral between the curves for each approach versus the control data. The legend indicates the number of grains ($n$) making up each curve. Note the reduction in x-axis scale between the columns, where the right, truncated distributions are plotted on a narrower range to emphasize the remaining discrepancies.

percentage misfits as lower errors at smaller percentiles (e.g., 0.2 $\psi$ precision at a grain size of 6.5 $\psi$ (91 mm) is a 13–15% misfit, whereas, a 0.01 $\psi$ precision at 4.5 $\psi$ (23 mm) is a 4–10% misfit).

### 6.4.3 Results: Handheld Image

As a final test for the KMS and AIF approaches, we turn towards our handheld imagery taken from S14A with a 4-times
5 improved resolution of 0.32 mm/pixel (Fig. 17). We only show the 20-pixel truncated results, which displayed high KS-test $p$-values > 0.2 and $A_{diff}$ close to 0 in both cases, with the AIF approach slightly underestimating ($A_{diff}$=0.6) and KMS slightly overestimating ($A_{diff}$=−0.77). For the KMS approach $m$ and $e$ are 0.07 and 0.05 $\psi$, respectively, and −0.06 and 0.05 $\psi$ for AIF.

### 6.5 Caveat of AIF

The promising results of the AIF approach shown in Figure 14–17 come with some consideration of the grain-by-grain accu-
10 racy. In Figure 18, we analyze the percentage of grains found in the AIF approach that have a corresponding grain in either the hand-clicked control (based on a 6-mm buffer of the b-axis line) or the KMS results (based on a 6-mm centroid buffer). From this subset of grains, we consider the AIF grain to be a matching (or correct) result if the b-axis difference between it and the nearby "good" grain (from the control or KMS) is < 1 cm. From this we see that in the best-case scenario the percentage of correct grains identified by the AIF approach is only 70%, from the handheld 0.32 mm/pixel image. A number of sites (S10,
S16, S20B, S24, S34, and S35) have < 50% matched grains. The two poorly performing sites (S24 with grain complexity and S34 with image blur) both demonstrate the lowest accuracy with < 40% matches. Notably, despite a significant number of false




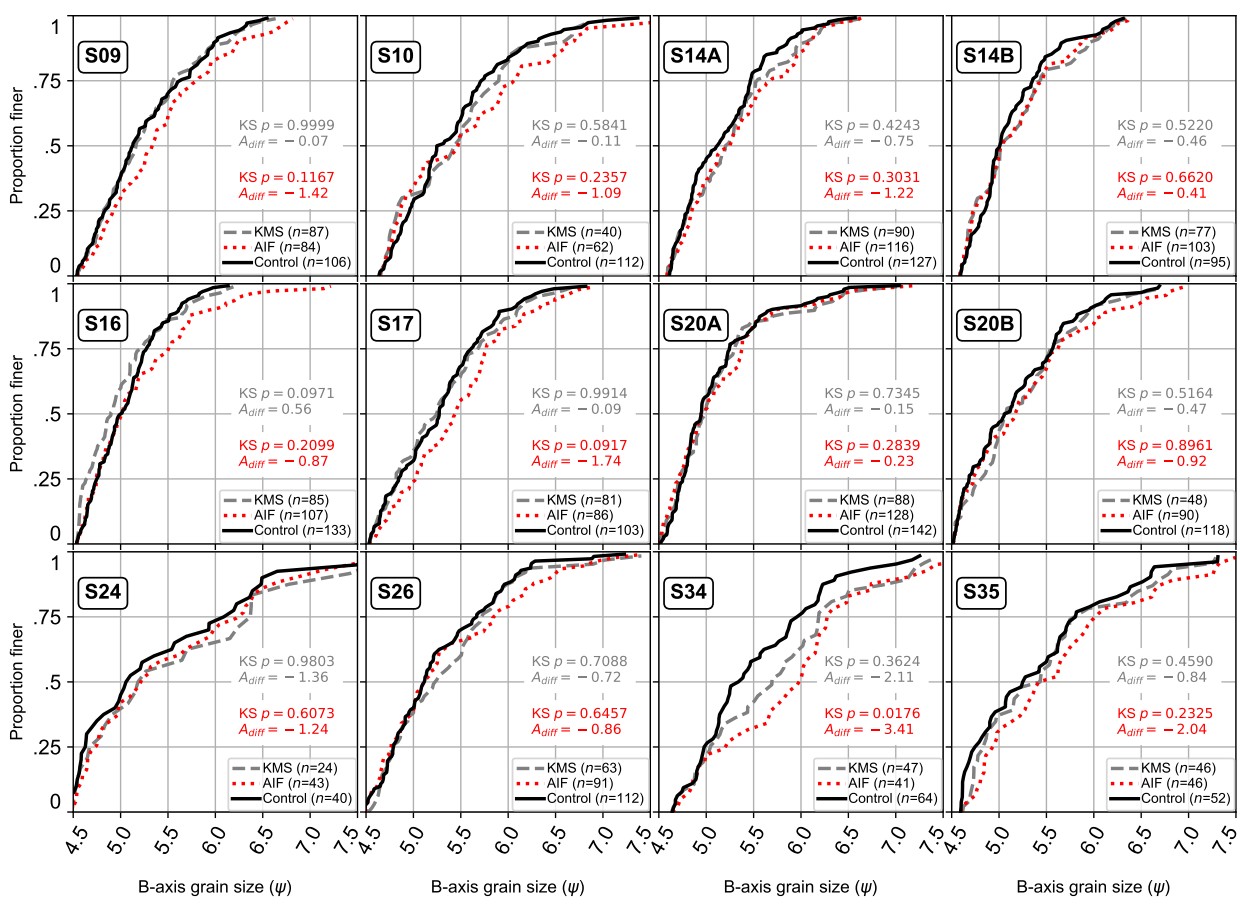

**Figure 15.** Comparison of 20-pixel truncated GSDs between hand-clicked control (black line), KMS *PebbleCounts* (gray, dashed line), and AIF *PebbleCountsAuto* (red, dotted line) for the 12 × ∼1.16 mm/pixel control sites. In corresponding colors are the *p*-value results of a KS-test and the $A_{diff}$ approximate integral between the curves for each approach versus the control data. The legend indicates the number of grains (*n*) making up each curve.



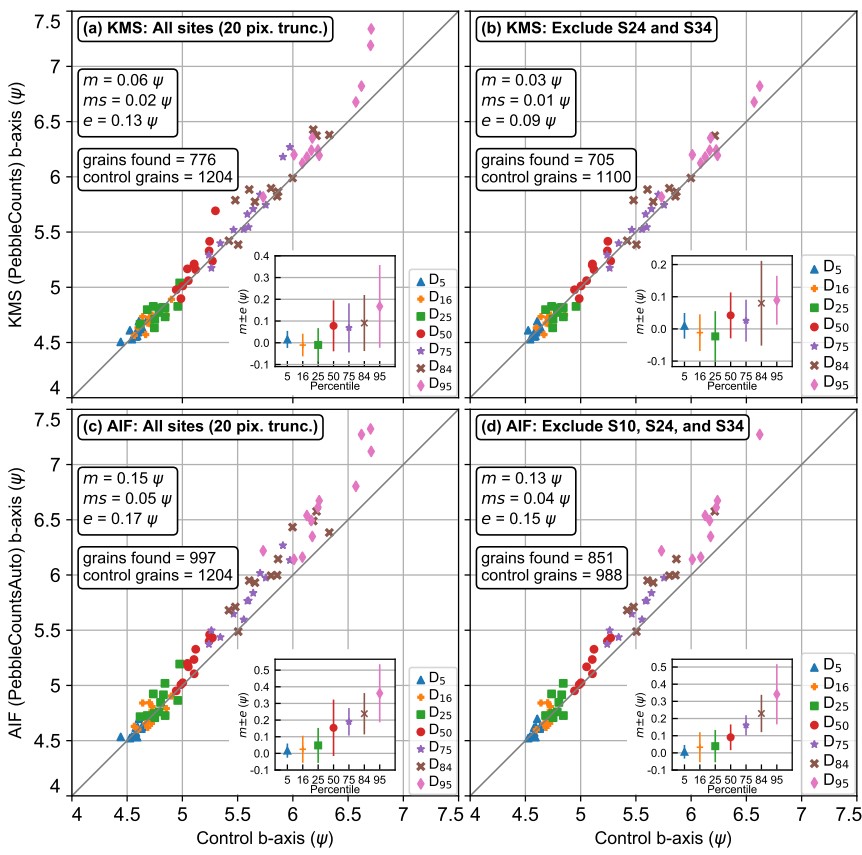

**Figure 16.** Comparing the key b-axis percentiles across all 12 field sites and between the KMS and AIF approaches with the 20-pixel truncation applied. (a) All 12 sites from KMS, (b) KMS improvement when excluding S24 and S34, (c) all 12 sites from AIF, and (d) AIF improvement when excluding S10, S24, and S34. For the main plot, each data point is a percentile value from a single site and the 1:1 relationship is the gray diagonal. The mean ($m$), mean squared ($ms$), and irreducible ($e$) errors are shown for each plot, taken as the average of all 7 percentile errors across the 9–12 sites plotted. The $m$ and $e$ are separately plotted for each percentile in the inset plot. The number of grains in the control ("control grains") and KMS or AIF results ("grains found") are also indicated.



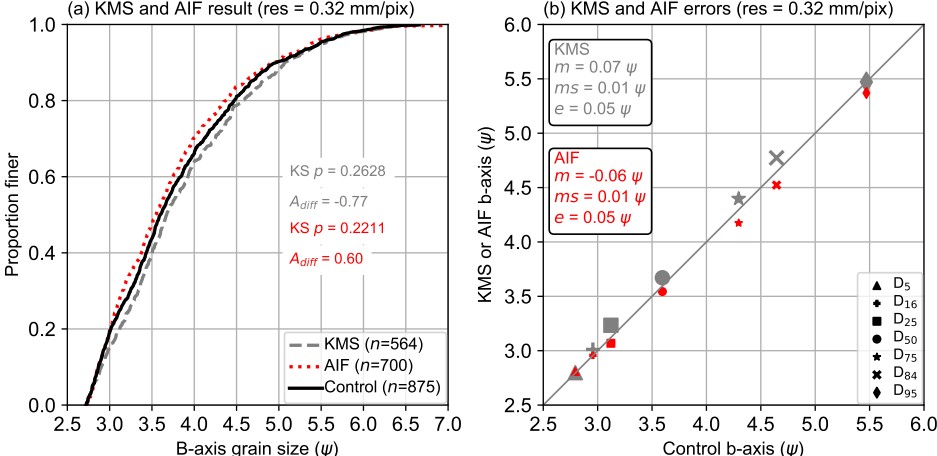

**Figure 17.** (a) Results from hand-clicked control (black line), KMS *PebbleCounts* (gray, dashed line), and AIF *PebbleCountsAuto* (red, dotted line) from the 20-pixel truncated run on the 0.32 mm/pixel handheld imagery. In corresponding colors are the *p*-value results of a KS-test and the $A_{diff}$ approximate integral between the curves for each approach versus the control data. (b) Percentile comparison for both methods with KMS in gray and AIF in red, with inset box showing the uncertainties for each in the corresponding color.

positives in the results, when comparing the overall GSDs (Fig. 14), and on a site-by-site basis (Fig. 15), the distribution of the AIF results matches the hand-clicked control well.

Figure 19 demonstrates the issues with the AIF approach in a few map-view examples of the results of the KMS approach versus the same pebbles in the AIF approach. On a grain-by-grain basis, there are many inaccuracies falling into three main

categories: over-segmentation of grains with internal edges and the selection of each segment as a separate grain, under-segmentation and merging of neighboring grains that have weak edges sometimes caused by image blur, and misidentification of non-grain objects or clusters of small grains. It is clear from this analysis that caution must be used when interpreting AIF results, particularly in complex or blurry images.

## 7  Discussion

### 7.1  Performance of KMS and AIF

Similar to other methods (Butler et al., 2001; Graham et al., 2010), the KMS *PebbleCounts* approach undercounts grain sizes in each respective size class. This undercounting does not undermine the resulting GSDs and associated percentile estimates, so long as an appropriate lower truncation is defined. This cutoff was found to be 20 pixels in b-axis length (Fig. 13), which explains the degradation in 3–5 mm counting in the reduced resolution lab images (Fig. 8), where the smallest pebbles were

only a few pixels in size as resolution was decreased.



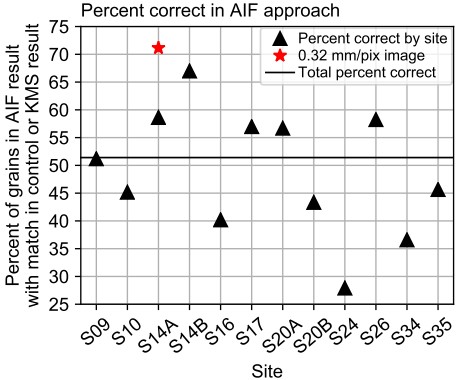

**Figure 18.** Percentage of grains from AIF results with a matching grain in either the hand-clicked control or in the KMS result. A match is defined as a grain within 5 pixels of the hand-clicked line or the KMS grain centroid for the 1.16 mm/pixel imagery, or within 20 pixels for the 0.32 mm/pixel image (corresponding in both cases to a distance of ~6 mm), and with a 1 cm maximum b-axis difference between the AIF grain and the match. The total percent correct, taken across all black triangles, is 51%.

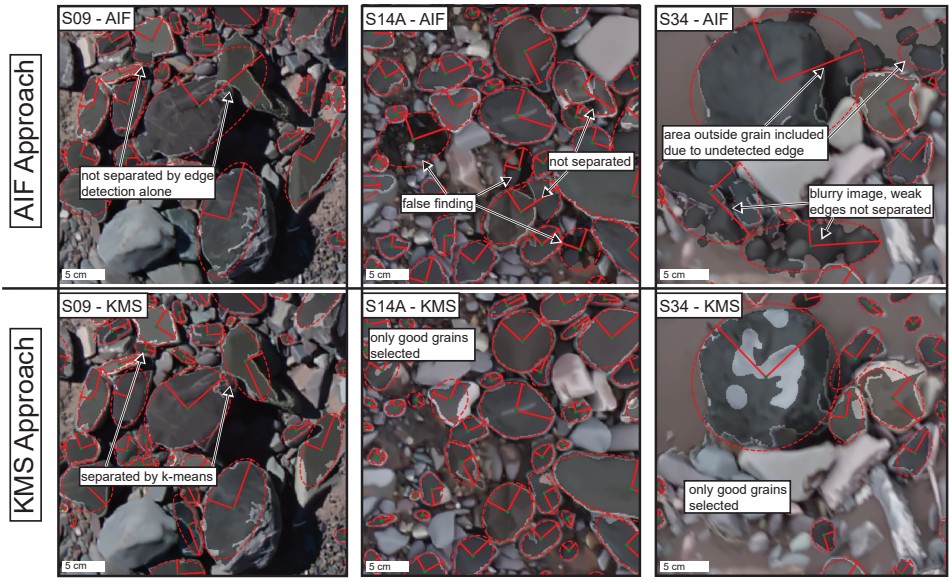

**Figure 19.** Resulting delineated grains using the AIF *PebbleCountsAuto* function (top row) versus the same area from the KMS *PebbleCounts* function (bottom row). Labels indicate the issues with the AIF results and improvement in KMS results. Note the poor results for the blurry image on the right (S34).



Applying this truncation (and excluding the poorly performing sites) to the KMS approach across 10 field sites with a total of 705 grains measured (versus 1100 in the control) results in $m$ and $e$ of 0.03 and 0.09 $\psi$, respectively, for the $\sim$1.16 mm/pixel imagery and 0.07 and 0.05 $\psi$ for the 0.32 mm/pixel image. For the AIF approach these values are 0.13 and 0.15 $\psi$ for the $\sim$1.16 mm/pixel imagery and $-0.06$ and 0.05 $\psi$ for the 0.32 mm/pixel image. These uncertainties are in the range of

previously published errors of 0.007–0.33 $\psi$ from similar techniques (Graham et al., 2010). Some studies make comparisons in mm rather than $\psi$ units, and, since the $\psi$ scale is logarithmic, the error in mm increases with $\psi$ from $\sim$0.8 mm uncertainty at 4.5 $\psi$ (23 mm) to $\sim$7 mm uncertainty at 6.5 $\psi$ (91 mm) for the $\sim$1.16 mm/pixel imagery in the KMS case. This is similar to previously reported uncertainties from *Basegrain* (Westoby et al., 2015) and better than the wavelet texture method applied to natural images (Buscombe, 2013). We also note that the uncertainties increase in $\psi$ at higher percentiles ($\geq D_{50}$), and we thus

suggest a higher error budget at higher percentiles.

As demonstrated in Figure 18 and 19, there are significant inaccuracies associated with the AIF approach. The errors associated with the AIF approach can be limited when up-scaling the automated function to cover large areas with tens-of-thousands of grains on high-quality (low-blur) $\sim$1 mm/pixel resolution imagery, with better results possible on < 0.5 mm/pixel imagery. However, to assess this error, it is recommended that users interested in applying the AIF *PebbleCountsAuto* tool to a large

survey site first apply the KMS *PebbleCounts* tool to a subset of the area, and use these results as a control for validation of the automation.

### 7.2    Effect of Lower Truncation on GSD

The issue of lower truncation on GSDs and percentile estimates has received much attention in the literature (e.g., Fripp and Diplas, 1993; Rice and Church, 1996; Bunte and Abt, 2001; Graham et al., 2010). Previously, field geomorphologists

were interested in all grains above 8–16 mm, simply because smaller grains were difficult to manually identify and thus underrepresented in the results (e.g., Fripp and Diplas, 1993; Rice and Church, 1998). Previous work suggests that truncation at the finer end of the distribution primarily increases the lower percentiles, while having less effect on the large ($> D_{50}$) percentiles (Bunte and Abt, 2001). We find significant shifts in all percentiles of $> 0.5$ $\psi$ when applying a 20-pixel truncation. Graham et al. (2010) report truncation errors of $< 0.3$ $\psi$ for all percentiles in 1, 3, and 5 $\psi$ truncated distributions. Their better

results at lower percentiles are likely because the data were collected manually grid-by-number style in the field with the ability to include smaller grain sizes. The measurement resolution presents the ultimate control on how accurately grain-size percentiles can be measured. The purpose of the KMS and AIF approaches introduced here is in acquiring GSDs from a subset of the full grain-size range present in the river, namely the subset with $> 20$-pixel b-axis length in image resolution.

### 7.3    Practical Considerations for Image Collection and Processing

Consistent with the results of other studies (e.g., Carbonneau et al., 2018) using orthometric versus top-down imagery, we find the difference in calculated resolution and subsequent GSDs to be negligible at these scales. While the use of orthomosaic imagery is not necessary, it may be preferable for capturing large sites at a constant resolution that can be tiled and fed into the algorithm.





The collection of 9+ photos as in our field surveys is not necessary for creating orthorectified images in *Agisoft*. As the textured nature of gravel images result in abundant match points, we were able to get comparable results in reduced time with only four photos, but overlap must be > 80% to ensure best results. Where a user desires accurate and dense point cloud data in addition to the 2D orthomosaics, it is recommended that (many) more images closer to the surface be collected and from oblique viewing angles (e.g., Verma and Bourke, 2019). In any case, the KMS *PebbleCounts* tool is recommended to be applied to maximum 1–2 m$^2$ patches, depending on the image resolution, as the manual clicking of good grains is time consuming. On the other hand, the AIF *PebbleCountsAuto* tool can theoretically be applied at larger scales, however, it is also advisable to tile data and feed it to the algorithm in maximum 1–2 m$^2$ patches for ~1 mm/pixel imagery, since the non-local means denoising takes a long time on very large images. Again, the usage of a GPU or large memory system will shorten processing times and allow for larger images to be run.

In terms of camera and photographic height considerations, one first needs to assess the minimum grain size that is desired. Following this, the resolution of the image can be determined using eq. (3) with some knowledge of the camera parameters (focal length in mm, camera height in mm, sensor size in mm, and image size in pixels). The smallest grain b-axis needed should be 20-times this resolution. For instance, using a similar camera to the Sony $\alpha$6000 (24 MP, 15.6×23.5 mm sensor, 16 mm focal length), to measure all grains down to 1 cm one needs a resolution of 0.5 mm/pixel, and thus a maximum camera height of ~2 m. In the case of a DJI Mavic drone with a 12 MP camera, wide angle 4.3 mm focal length, and 4.55×6.17 mm sensor, this 0.5 mm/pixel resolution requires an unreasonably low flight height of ~1.4 m, giving a field of view of only ~1.5×2 m. If finer grain sizes are desired, the user can use higher resolution imagery but must be aware of the longer time needed for processing < 0.5 mm/pixel imagery.

## 7.4 Additional Data Dimensions from Point Clouds

The results presented here are similar to other studies segmenting grains from 2D imagery. This ignores the potential to exploit the third height dimension of the data from irregularly spaced SfM-MVS point clouds and associated DEMs. Many authors have already begun to look at patch-scale variance or roughness (e.g., Rychkov et al., 2012) from point clouds on gravel-bed rivers to determine bulk characteristics, but this stops short of object detection and segmentation. Here, we briefly describe some of our own efforts to incorporate this additional information into *PebbleCounts*.

Our simplest approach was including the gridded DEM information, resampled to the same resolution as the orthomosaic. We inverted the elevation raster and flood-filled from the lowest points (tallest grains) using watershed approaches, conceptionally similar to lidar tree-detection algorithms (e.g., Chen et al., 2006; Alonzo et al., 2015). For large, prominent grains with semi-spherical shapes, the flooded area was found to linearly increase until reaching the grain boundary, at which point the rate of area change jumped. We explored this break point as a potential segmentation tool for larger grains, but found that in the complex natural setting the shape of most grains is far from spherical, and furthermore, overlapping grains led to inconsistent behavior in the area breaks.

In an additional approach, we calculated both roughness and curvature at a variety of scales (5, 10, 50, 100 mm) directly from the point cloud using the open-source *CloudCompare* software (CloudCompare, 2018). This information was then gridded





into a raster of the same resolution of the orthomosaic. While roughness could at times identify the smoother sand patches, it was difficult to discern between a sand patch and flat rock, and a color threshold on the orthoimagery was more successful. Curvature showed some spikes at grain boundaries, with the potential to aid in edge detection, however, we found that curvature was also high on intra-granular features.

In general, this analysis was complicated by vertical noise (scattering around a mean value) inherent to the SfM-MVS technique in the generation of dense point cloud data. In the field, for ∼9 photos taken from a height of ∼5 m, the vertical standard deviation of points on a detrended flat surface (one of our coded targets) was found to be 1.7 mm for 13,014 points. On the other hand, in the perfect lab setting with 16 photos from ∼1.5 m, the detrended flat carpet around the pebbles achieved a standard deviation of 0.2 mm (33,371 points), similar to other SfM-MVS studies using large numbers of carefully collected
images (e.g., Cullen et al., 2018; Verma and Bourke, 2019). These standard deviations from detrended flat surfaces represent a best-case scenario, whereas, in our field setting, the vertical uncertainty on the complex, overlapping pebbles is likely higher. Such vertical noise is absent from the orthomosaics and limits the applicability of point clouds at these scales.

Ultimately, as the point cloud actually has a lower resolution (since it is based only on matched points) and more vertical noise than the orthomosaic (which exploits the full camera resolution), the imagery alone provided more detail. This is par-
ticularly important around grain edges needed for segmentation, which are not captured in top-down imagery alone, as shown in Figure 20. The lab setting resulted in point clouds with sufficient density and precision to identify individual grains with point-cloud processing tools. Thus, achieving higher quality SfM-MVS point clouds is possible, but only through more intense data collection during fieldwork (Fig. 20).

Alternatively, lidar point clouds with distance measurements based on phase shifts have a lower standard deviation of ∼1
mm in multiple settings and distances (up to ∼300 m) and could allow more precise delineation using roughness and curvature calculations directly on the point cloud, however, such devices remain costly. Additionally, the development of affordable hyperspectral cameras with additional wavelengths will help in image segmentation in the spectral domain. To conclude, the potential for additional data dimension integration into pebble counting may be possible using higher dimensional object detection schemes, but, for the time-being, the orthoimagery alone provides satisfying results.

## 8   Conclusions

Using a k-means approach for pebble segmentation in the spectral and spatial domain combined with fast manual selection of good results, we developed a new semi-automated algorithm for grain sizing optimized for images taken over gravel-bed rivers (*PebbleCounts*). We also developed an automated algorithm that uses suspect grain filtering (*PebbleCountsAuto*), albeit with larger uncertainties in the results. The lower truncation of the methods (minimum b-axis length measurable) is limited
to 20-pixels and above. These new methods were necessary to acquire grain-size distributions from dynamic high-mountain rivers with complexity from sources such as large ranges in grain size, intra-granular heterogeneity, grain overlap, irregular shadowing, and sand patches. Similar to previous methods, *PebbleCounts* is best applied at the patch scale (1–10 m$^2$), however, *PebbleCounts* provides more realistic results in complex images without any post-processing steps in ∼5 minutes per patch,



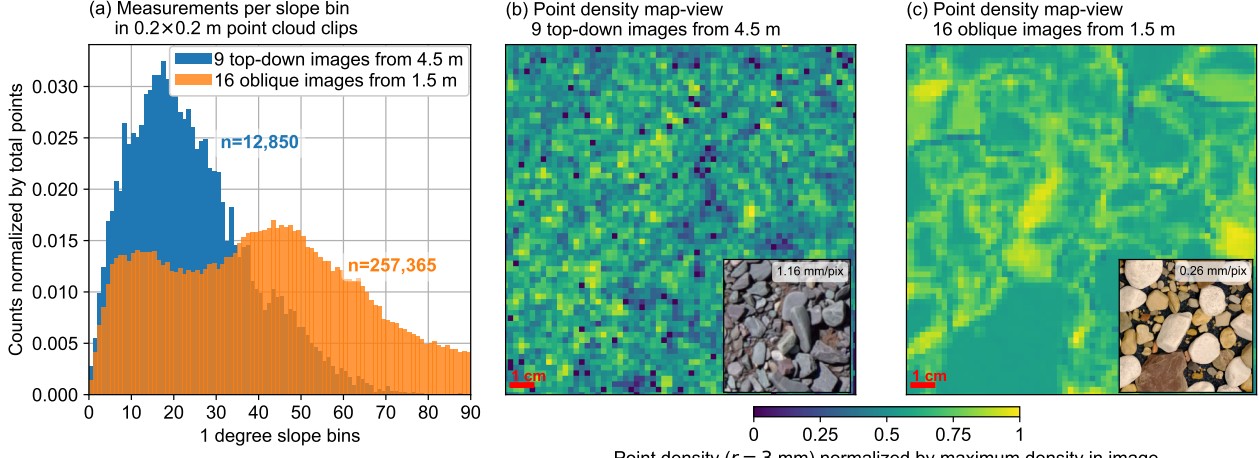

**Figure 20.** (a) Slope distribution in field (top-down) and experimental (oblique) point cloud clips. The point cloud slope was calculated in *CloudCompare* (CloudCompare, 2018) by first calculating the normals at each point using the 6 nearest neighbors and then extracting the dip of each normal. (b) Map-view of point density normalized by the maximum for the 9 top-down field images and (c) the same for the 16 oblique experimental images. Point density was calculated as the number of points in a radius of 3 mm. The clips were from a 0.2×0.2 m area, visually selected to have similar grain sizes and numbers of grains, shown in the inset images in (b) and (c). The average point density for the 16 oblique photo setting was 59 points/cm$^2$, whereas, in the field using 9 top-down photos the density was 17 points/cm$^2$. Note the higher point density on grain edges in (c) compared to (b), which are improtant for segmenting grains directly on the point cloud.

assuming ∼1 mm/pixel resolution imagery. *PebbleCountsAuto* performs very well on high-quality (low-blur) imagery, though with remaining misidentification that must be approached with caution. Grain-sizing results can be upscaled to areas on the order of $10^2$–$10^4$ m$^2$ when *PebbleCounts* results are used as calibration and validation for the automated *PebbleCountsAuto* function. Such areas can be readily surveyed at ∼1 mm/pixel resolution with the 12–24 MP cameras found on many drones and consumer cameras, presenting the potential for the generation of full grain-size distribution maps at the scale of entire river cross sections and over shorter reaches.

*Code availability.* *PebbleCounts* is a Python based program with the code and documentation available on GitHub at: https://github.com/UP-RS-ESP/PebbleCounts (Purinton and Bookhagen, 2019).

*Author contributions.* BB and BP defined the project. BP developed the algorithms with support from BB. BP carried out the analysis, produced the figures, and wrote the manuscript. BB provided funding, guidance in data analysis, and manuscript edits.

 

*Competing interests.* The authors declare that they have no conflict of interest.

*Acknowledgements.* Anna Rosner is thanked for assistance with fieldwork for mast surveys. Steffen Wellegehausen is thanked for aiding in the lab experiment setup. Funding was sourced from DFG Funded IRTG-StRATEGy (IGK2018) and NEXUS funded through the MWFK Brandenburg, Germany, both for Bodo Bookhagen. We acknowledge the support of the Open Access Publishing Fund of the University of Potsdam.



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
