# Peer review of "Introducing PebbleCounts: A grain-sizing tool for photo surveys of dynamic gravel-bed rivers"

_Earth Surface Dynamics, 2019_

## Referee Comment (RC1) · Patrice Carbonneau (Referee) · 16 May 2019

This is an excellent contribution which comes at timely point when many technologies are coming together. The new method rests on a genuine innovation in grain size mapping, a clever use of a k-means cluster. The paper is mostly well written and has an excellent level of technical detail. This might be demanding for readers, but a detailed reading of this work is effective in lifting the black-box effect that can arise from advanced image processing workflows with a large number of steps and parameters. One of the major benefits of this work is the fact that it is open-source and written in the popular Python language. This is very timely because Basegrain, the current best

option for grain size mapping, is written for windows-7 and no longer in active development. Despite the fact that the creator of Basegrain is very collaborative and willingly shares the Matlab source code, the river sciences community will soon need an updated option with a major preference towards an open-source solution. PebbelCounts seems poised to fill this gap.

I have a few suggestions about corrections but these are not major:

1- I found equations 1,2 and 3 to be laborious and not really necessary. Non-metric RGB cameras destined for the consumer or 'prosumer' market have square pixels. As observed by the authors, differences in X and Y resolution are negligible. This whole section could be cut short to a single equation.

2- A better reading of your bibliography. This bibliography is in fact quite complete. But I get a distinct impression that many papers were skimmed, deemed relevant, and cited. I am often struck by points of discussion or relevant findings of other authors which are missing in the text despite the fact that these authors are cited. Another explanation may be that the discussion lacks many key points of a good discussion where elements of other authors will need more consideration. Specifics of this will be seen below.

3- The accepted term for 'top-down' imagery in the remote sensing community is 'nadir'. Please use that term.

4- The drone/SfM elements of the paper are not well organised. The UAV/SfM paragraph in the introduction could be moved. SfM is now so ubiquitous that it should not shock the reader if you say in the methods that SfM was used for data acquisition. The overall reflection of how this could apply to drones in an SfM workflow needs to be moved to a section in the discussion. This is an area where many elements of the cited literature are not mentioned. Important points are: i- Acquisition geometry. There is now a large volume of literature on the image geometry that produces the best 3D models from SfM. This remains important here since DEM- distortions will propagate to the orthoimage. So in parallel with the robotic photosieving work, there should be a rec-
ommendation of a mixed acquisition with nadir imagery for the actual grain delineation but with oblique views for maximum SfM quality.

ii- Be honest and realistic about scale coverage. The paper states an ambition of covering areas up to 10 000 m2. At the same time the method rests on SfM with surveyed ground control to generate an orthomosaic with a constant resolution. This is in fact an ambitious goal. The acquisition of 80% overlapping imagery with surveyed GCPs and at sub-mm resolutions over a hectare is a multi-day (or multi-camera) job. This is why the robotic photosieving approach of Carbonneau et al (2018) does not advocate a orthomosaics but uses scaled individual images.

iii- Use of a Mavic as the only reference is perhaps overly pessimistic. Carbonneau et al (2018) mention that a Phantom 4 Pro (20 Mpix imagery) could acquire 0.7mm/pix imagery at 2m altitude. With active collision avoidance, that becomes a workable, if very low, flying altitude. This problem should resolve itself as sensors with more than 20MPix become more available.

5- Improve the discussion. The discussion needs a much improved start and overall re-organisation. A good rule on writing a discussion is to start with a sentence or two that distil the major findings that you want the reader to take away from this paper. A discussion needs to go over the substantive elements of the findings and their meaning and contrast to the work of other authors before going into issues. Here, the authors need to start the discussion with a section that tells us what they have achieved and gives the reader a better sense of how PebbelCounts compares to other methods. Without necessarily running other methods on their data, we at least need a summary table that presents errors reported in literature and compares them to the current work. This is the section where you need to show an enhanced understanding of the literature.

6- 3D data The section at the end of the discussion on the integration of 3D information does not sit well. Move it to the introduction with the intention of stating that you will not be using 3D clouds or DEMs. It would be worth citing the work of James Brassington

and Damia Vericat who have developed particle sizing based on TLS. But also you could mention that Woodget et al (2018), already cited, found that 3D information did not improve particle size estimates. This section could be the place in the intro where you discreetly place a few SfM/drone citations but just to further justify that this will be an image-based method

These minor corrections should make the paper ready for publication.

Patrice Carbonneau May 2019

---

## Referee Comment (RC2) · Pascal Allemand (Referee) · 30 May 2019

Review of the paper entitled: " Introducing PebbleCounts: A grain-sizing tool for photo surveys of dynamic gravel-bed rivers" by Benjamin Purinton and Bodo Bookhagen

This paper describes an interesting open source software for automatic measurement of grain size distribution from images. Compared to existing systems, this open source software is able to work on ortho-images obtained by phogrammetric methods on image collections covering wide areas. The algorithm seems very efficient but the results are deserved by the text which is too long and a discussion where key problems are downed out among less important elements. I suggest to the authors to re-write the

paper in a more concise and linear way. P4 line 2: How to be sure that the detected grains are representative of the whole grains? It is a point to discuss. P4 line 6 : The fact that current methods are limited to some m2 is a real limitation that should be indicated in the part concerning the current methods. Figure 1: the differences in results between AIF KMF and water shed methods should be discussed in the discussion Figure1: Concerning the watershed method (basaGrain ?), you show, I thing, the gross results. The results can be filtered with basegrain by post processing. Page 5 line8 : What type of denoising method do you use ? Does it preserve edges ? Page 5 line 10: how do you filter the sand patches ? on color ? on texture ? Figure 3: I suggest to merge figure 3 and figure 6 and to shorten the text referring to the manual user of your software. Figures 4 and 5: difficult to read and not necessary. I suggest to remove or to rework in a more concise and readable way Concerning "5 Calibration and Validation Test I: Controlled Experiment": shorten and get to the point. The part concerning the cameras is not useful. What is important is the result (description of the Photoscan parameters is useless for example). Size of pixels do not matter. What is important is the ratio between the resolution of the image to the size of the smallest grain detected. Same for "6 Calibration and Validation Test II: Field Surveys". I suggest to remove the useless details and to go to the point. You could show only the better and the worth examples and discuss why the "best" example give good results and why the "worst" example give no such good results (but good anyway ïĄŁ ) Figure 19 (they are too many figures): for me, what is important is to discuss why its work or not, in what case and How I can use your software and what error can I expect, by adding some advices on the acquisition procedure I should follow. These points are discussed in the current version but are not enough highlighted.

---

## Author Comment (AC1) · 13 Jun 2019

Reply to reviewers for manuscript (esurf-2019-20) submission to Earth Surface Dynamics:

Introducing PebbleCounts: A grain-sizing tool for photo surveys of dynamic gravel-bed rivers – Purinton and Bookhagen

Highlighted in **bold** are the reviewer comments followed by our point-by-point replies in regular text. Sentences that will be added or changed from the original manuscript are in *italics*. All changes will be made to the final manuscript submission following completion of the interactive review period.

**Reviewer Patrice Carbonneau**

**This is an excellent contribution which comes at timely point when many technologies are coming together. The new method rests on a genuine innovation in grain size mapping, a clever use of a k-means cluster. The paper is mostly well written and has an excellent level of technical detail. This might be demanding for readers, but a detailed reading of this work is effective in lifting the black-box effect that can arise from advanced image processing workflows with a large number of steps and parameters. One of the major benefits of this work is the fact that it is open-source and written in the popular Python language. This is very timely because Basegrain, the current best option for grain size mapping, is written for windows-7 and no longer in active development. Despite the fact that the creator of Basegrain is very collaborative and willingly shares the Matlab source code, the river sciences community will soon need an updated option with a major preference towards an open-source solution. PebbelCounts seems poised to fill this gap.**

We thank the reviewer for positive comments regarding the quality, thoroughness, timeliness, and innovativeness of the submission. The level of detail in the paper is very high, which this reviewer found good, but the second reviewer found obfuscating to our main points. Our goal with the manuscript at this level of detail was to provide sufficient information for a reproducible algorithm and processing chain that can be precisely followed. Previous publications of grain-size estimation algorithms often had a shorter method section that did not allow to reproduce the algorithm. In the spirit of open source and traceable and reproducible algorithms and software, we strongly think that a well-documented algorithm is beneficial to the community for the years to come. We cover points regarding this more in the reply to the second reviewer, but are happy to see that the first reviewer found the highly detailed analysis and writing useful.

**I have a few suggestions about corrections but these are not major:**

**1- I found equations 1,2 and 3 to be laborious and not really necessary. Non-metric RGB cameras destined for the consumer or 'prosumer' market have square pixels. As observed by the authors, differences in X and Y resolution are negligible. This whole section could be cut short to a single equation.**

We agree that this section can be cut-down without losing too much information, but still feel a description for the interested user warrants a few sentences. Section 5.3.1 will now read (including the suggestion in point 3 below):

*We refer to all imagery used as top-down as opposed to the commonly used nadir term, which refers to images taken consistently from a directly downward-pointing vantage, since our images are taken from a variety of near-downward angles. As consumer-grade cameras have square pixels with negligible difference in horizontal and vertical resolution, the image scale can be calculated directly from the camera parameters and camera height with the resolution (R) in mm/pixel given by:*

$R = (S * h) / (f * I)$ (1)

*where S is the sensor height or width in mm, f is the lens focal length in mm, h is the camera height in mm, and I is the image height or width in pixels. S and I should either both be the width, or both be the height of the sensor and image, respectively. This assumes no major distortions within the field of view, which is not valid for oblique imagery, but is negligible for top-down photography at close range using non-fisheye lenses. With h=1.55 m, the resulting image resolutions tested from the Fujifilm were 0.26, 0.35, 0.53, and 1.05 mm/pixel by eq. (1).*

**2- A better reading of your bibliography. This bibliography is in fact quite complete. But I get a distinct impression that many papers were skimmed, deemed relevant, and cited. I am often struck by points of discussion or relevant findings of other authors which are missing in the text despite the fact that these authors are cited. Another explanation may be that the discussion lacks many key points of a good discussion where elements of other authors will need more consideration. Specifics of this will be seen below.**

We have tried to fix and improve these points through some re-reading of the relevant bibliography, and re-writing of discussion points. We have made an effort to avoid spending much time discussing grain-sizing efforts that are based on texture (roughness, variance, entropy) techniques, besides to mention the papers that cover them, which are many in number and highly variable in exact methodology. We have focused our closer readings and the discussion on those studies that also employ image-segmentation techniques to the problem of grain sizing, as these are more relevant to our study and allow to decipher the full grain-size spectrum.

**3- The accepted term for 'top-down' imagery in the remote sensing community is 'nadir'. Please use that term.**

This point is well taken and we have discussed this among the two authors. Nadir imagery refers specifically to downward-pointing images. The images used in this study are taken from a variety of angles near downward, but hardly ever exactly, since we are using an imperfect camera-on-mast setup. Top-down imagery is a more general term and we feel more appropriate

for the type of imagery used in this study, especially because we do not measure the angle of the images taken. We have, however, added a sentence at the beginning of Section 5.3.1 on P10, L5:

*We refer to all imagery used as top-down as opposed to the commonly used nadir term, which refers to images taken consistently from a directly downward-pointing vantage, since our images are taken from a variety of near-downward angles.*

**4- The drone/SfM elements of the paper are not well organised. The UAV/SfM paragraph in the introduction could be moved. SfM is now so ubiquitous that it should not shock the reader if you say in the methods that SfM was used for data acquisition. The overall reflection of how this could apply to drones in an SfM workflow needs to be moved to a section in the discussion. This is an area where many elements of the cited literature are not mentioned. Important points are:**

The paragraph in the introduction concerning UAV and SfM orthomosaic generation can be shortened, but it is mainly there to provide a segue from previous manual counting methods in the previous paragraph, to photo-sieving methods, and onward to the last introductory paragraph where we introduce our photo-sieving method. We will shorten the paragraph as follows, and will create a new section in the discussion to cover the other concerns of the reviewer (see the next reviewer points below). The new introduction paragraph from P2, L10:

*In light of this, measurement from photographs is an attractive option for increasing sample size and decreasing fieldwork, while covering larger areas. Increasingly affordable high-resolution --- 12--24 megapixel (MP) --- cameras, allows the collection of high-quality photo surveys at scales of entire river cross sections or reaches via Structure-from-motion with Multi-View Stereo (Smith et al., 2015; Eltner et al., 2016) at resolutions at or exceeding 1 cm/pixel (e.g., Woodget and Austrums, 2017). Even higher resolution (1 mm/pixel) river surveys can be accomplished with low-flying unmanned aerial vehicles (UAVs) (e.g., Carbonneau et al., 2018), pole-mounted cameras, or using handheld imagery.*

**i- Acquisition geometry. There is now a large volume of literature on the image geometry that produces the best 3D models from SfM. This remains important here since DEM-distortions will propagate to the orthoimage. So in parallel with the robotic photosieving work, there should be a recommendation of a mixed acquisition with nadir imagery for the actual grain delineation but with oblique views for maximum SfM quality.**

**ii- Be honest and realistic about scale coverage. The paper states an ambition of covering areas up to 10 000 m2. At the same time the method rests on SfM with surveyed ground control to generate an orthomosaic with a constant resolution. This is in fact an ambitious goal. The acquisition of 80% overlapping imagery with surveyed GCPs and at sub-mm**

**resolutions over a hectare is a multi-day (or multi-camera) job. This is why the robotic photosieving approach of Carbonneau et al (2018) does not advocate a orthomosaics but uses scaled individual images.**

**iii- Use of a Mavic as the only reference is perhaps overly pessimistic. Carbonneau et al (2018) mention that a Phantom 4 Pro (20 Mpix imagery) could acquire 0.7mm/pix imagery at 2m altitude. With active collision avoidance, that becomes a workable, if very low, flying altitude. This problem should resolve itself as sensors with more than 20MPix become more available.**

Regarding the points i-iii listed here, we have re-written Section 7.3 in the discussion. This section deals with practical considerations for image collection and processing using the proposed algorithms. We now include sub-sections dealing with image resolution and geometric considerations for image collection, using a UAV-SfM robotic photosieving workflow (as covered by Carbonneau at al. (2018)), and addressing the ambitious up-scaling we propose. This new section hopefully addresses the reviewers concerns and adds clarity to the potentials and caveats of PebbleCounts as applied to photogrammetric river surveys, particularly regarding the use of drones. The new Section 7.3 in the discussion will read as follows (beginning from the current location on P25, L29):

[revised manuscript text omitted]

**5- Improve the discussion. The discussion needs a much improved start and overall reorganisation. A good rule on writing a discussion is to start with a sentence or two that distil the major findings that you want the reader to take away from this paper. A discussion needs to go over the substantive elements of the findings and their meaning and contrast to the work of other authors before going into issues. Here, the authors need to start the discussion with a section that tells us what they have achieved and gives the reader a better sense of how PebbelCounts compares to other methods. Without necessarily**

**running other methods on their data, we at least need a summary table that presents errors reported in literature and compares them to the current work. This is the section where you need to show an enhanced understanding of the literature.**

While we agree that the discussion could use some reorganization (see for example our response to point 4 i-iii above) and a better introduction, we would like to avoid tables with exhaustive lists of the errors reported in other studies. In particular, we do not want to make comparisons between errors from the described image segmentation approach and texture based (e.g., roughness, entropy, semivariance) approaches, since these other methods are based on correlative relationships, rather than direct measurements of the grains.

A comparison with other image segmentation studies is made especially complicated by the tendency in different studies to sometimes only report the bias without the error spread, use different metrics of uncertainty reporting, and/or report uncertainties in mm rather than psi units. For example, unfortunately, the only study we found that reports the accuracy of Basegrain versus control data is that of Westoby et al. (2015), however, they only provide percentile bias numbers in mm with no measure of spread.

We feel it is more useful to report a few aggregate numbers from these studies, which together demonstrate the PebbleCounts algorithm to be within the range (and even on the low-end) of previously reported uncertainties. Ultimately, the uncertainty in measurement is highly dependent on the input image quality and complexity (range in grain size, angularity, intra-granular variability) and providing blanket estimates is less useful than end-users applying the KMS tool to a subset of images to validate the results of the AIF approach.

To clarify these points and provide a better intro for the discussion, we propose a modification of Section 7 and 7.1, which begins at P23, L6:

*7 Discussion*

*In this study we developed two new methods for grain-size measurement with low uncertainties and the potential to deliver full GSDs from complex images of high-energy mountain rivers. Our open-source Python-based algorithms perform equally well to other image segmentation tools, but can be applied more quickly over larger areas surveyed by the SfM-MVS workflow we present. Critical to success is the application of a strict lower cutoff, which limits the minimum measurable b-axis grain size to 20-times the pixel resolution. The automated version of the algorithm delivers less accurate measurements, but these can be limited by using low-blur, higher resolution imagery. We focus our discussion on the comparison of our approach with similar work, the effect of the lower truncation on GSD estimates, and practical guidelines for acquiring imagery and applying PebbleCounts, including the application of UAV surveys.*

*7.1 Performance of KMS and AIF*

*For comparison of our algorithms to previous work, we do not consider errors reported in studies using texture-based measurements (e.g., Woodget at al., 2018), since these methods are based on correlative relationships rather than physical measurement of each grain. Similar to other image segmentation methods (Butler et al., 2001; Graham et al., 2010), the KMS PebbleCounts approach undercounts grain sizes in each respective size class. This undercounting does not undermine the resulting GSDs and associated percentile estimates, so long as an appropriate lower truncation is defined. This cutoff was found to be 20 pixels (compare to 23 pixels found by Graham et al. (2005a)) in b-axis length (Fig. 13), which explains the degradation in 3--5 mm counting in the reduced resolution lab images (Fig. 8), where the smallest pebbles were only a few pixels in size as resolution was decreased.*

*As shown in Figure 16, when we apply this cutoff and exclude poorly performing images we find an average m (bias) and e (spread) of 0.03 and 0.09 psi, respectively, for the ~1.16 mm/pixel imagery and 0.07 and 0.05 psi for the 0.32 mm/pixel image. For the AIF approach these values are 0.13 and 0.15 psi for the ~1.16 mm/pixel imagery and -0.06 and 0.05 psi for the 0.32 mm/pixel image. These are averages, which actually increase at higher percentiles in agreement with other image segmentation methods (e.g., Sime and Ferguson, 2003). We thus suggest higher error budgets at higher percentiles.*

*As demonstrated in Figures 18 and 19, there are significant inaccuracies associated with the AIF approach. The errors associated with the AIF approach can be limited when applied to high-quality (low-blur) ~1 mm/pixel resolution imagery, with better results possible on < 0.5 mm/pixel imagery. Ultimately, the uncertainties are highly dependent on the input image quality and complexity (range in grain size, angularity, intra-granular variability) and providing blanket estimates is less useful than end-users applying the KMS tool to a subset of images to validate the results of the AIF approach.*

*In spite of this caveat, our bias values of 0.03-0.13 psi are in the range of previously published absolute biases of 0.007--0.33 psi from similar techniques (see Table 2 in Graham et al. (2010)). To our knowledge, the only study to compare Basegrain results to control data by Westoby et al. (2015), makes comparisons in mm rather than psi units. Since the psi scale is logarithmic, in our study the error in mm increases with psi from ~0.8 mm uncertainty at 4.5 psi (23 mm) to ~7 mm uncertainty at 6.5 psi (91 mm) for the ~1.16 mm/pixel imagery in the KMS case. Westoby et al. (2015) report similar bias from Basegrain, again increasing in magnitude at higher percentiles. Regarding the error spread reported in the literature, our range of 0.05-0.13 psi is less than the 0.25 and 0.14 psi values reported by Sime and Ferguson (2003) and Graham et al. (2005b), respectively, for their image segmentation techniques.*

**6- 3D data The section at the end of the discussion on the integration of 3D information does not sit well. Move it to the introduction with the intention of stating that you will not be using 3D clouds or DEMs. It would be worth citing the work of James Brassington and Damia Vericat who have developed particle sizing based on TLS. But also you could mention that Woodget et al (2018), already cited, found that 3D information did not**

**improve particle size estimates. This section could be the place in the intro where you discreetly place a few SfM/drone citations but just to further justify that this will be an image-based method.**

We have significantly shortened this section on point clouds to a few key points and moved the majority of the information, including the current Figure 20, to the supplementary material. The remaining section is now in the introduction. We have incorporated the suggested citations. We have created a new section in the introduction preceding the current Section 4 where the PebbleCounts algorithms are presented. This will be the new Section 4 and reads as follows (we also include the new Section S1 below):

[revised manuscript text omitted]

**Reviewer Pascal Allemand**

**This paper describes an interesting open source software for automatic measurement of grain size distribution from images. Compared to existing systems, this open source software is able to work on ortho-images obtained by phogrammetric methods on image collections covering wide areas. The algorithm seems very efficient but the results are deserved by the text which is too long and a discussion where key problems are downed out among less important elements. I suggest to the authors to re-write the paper in a more concise and linear way.**

We appreciate that the reviewer finds the work relevant and sees its potential for grain-size mapping. However, we disagree that the paper needs major rewriting. As mentioned in our first reply, our goal in having such a high level of detail is in reporting every step in a very complex study. Interested users will be able to follow every step we have taken from data collection to algorithm development and assessment via a close reading. Hopefully some of the confusion was eliminated in the large amount of discussion rewriting and reorganization that we accomplished in reply to the first reviewer. Nevertheless, we have tried to accommodate the comments of the second reviewer in some of the points below.

**P4 line 2: How to be sure that the detected grains are representative of the whole grains? It is a point to discuss.**

We have hand-clicked every visible grain in the control images, thus representing the true distribution of the grains. We can be sure that the grains detected using the image segmentation approaches are representative of the whole distribution because these grain-size distributions match very well to the hand-clicked control data as shown in, for example, Figure 14. We add a sentence at P4, L2:

*Despite the selection of fewer grains, Figure 2 demonstrates that these grains do represent the entire distribution through the close match in GSD between hand-clicked and KMS results.*

**P4 line 6: The fact that current methods are limited to some m2 is a real limitation that should be indicated in the part concerning the current methods.**

We clarify this point at the start of Section 3 by modifying the first paragraph beginning at P3, L24:

*Watershed segmentation is effective for interlocking, uniformly colored, oblate grains, however, energetic gravel-bed rivers in mountains often have more complex grain compositions with intra-granular variation, irregular shadowing, and a large range of sizes. The automated watershed methods proposed suffer from over-segmentation, grain misidentification, and the need for significant, time-consuming post-processing (e.g., in Basegrain with the split, merge, and delete tools) when applied to complex images. These issues limit the application of previous methods at areas > 10 $m^2$.*

**Figure 1: the differences in results between AIF KMF and water shed methods should be discussed in the discussion**

We feel that this discussion belongs in the introduction. Our algorithm does not use the watershed method and this section and Figure 1 and 2 are intended to highlight the reason why we avoid this technique before we go on to explain the steps our algorithm does take, which is

certainly introduction material. We have however, added a sentence to the end of Section 7.1 in the discussion:

*Importantly, we emphasize that the previous image segmentation techniques discussed here all rely on the watershed segmentation step, whereas, neither of our algorithms use this step for the reasons demonstrated in Figures 1 and 2.*

**Figure1: Concerning the watershed method (basegrain?), you show, I thing, the gross results. The results can be filtered with basegrain by post processing.**

Yes, the results can be post-processed, but the point here is that the post-processing is time-consuming, subjective, and limits the applicability of Basegrain to larger areas. We have highlighted this point with the modified paragraph mentioned in the P4, L6 point above.

**Page 5 line 8: What type of denoising method do you use? Does it preserve edges?**

This is covered at P5, L12-13:

*...chromaticity bands from this color space undergo bilateral filtering (Tomasi and Manduchi, 1998) to preserve inter-granular edges while further smoothing color.*

**Page 5 line 10: how do you filter the sand patches? on color? on texture?**

This is stated at P5, L10:

*...HSV color selection for sand-patch masking.*

We clarify by adding:

*...HSV color selection for sand-patch masking (whereby sand is filtered by a narrow, user-selected color mask).*

**Figure 3: I suggest to merge figure 3 and figure 6 and to shorten the text referring to the manual user of your software.**

We feel that the text is sufficient and should not be cut for the sake of clarity to the user. We will merge Figure 3 and 6 as shown here:

[Figure]

*Figure 3. Flowchart of PebbleCounts (left) and PebbleCountsAuto (right). The boxes are user supplied input or output from the algorithm. Dashed lines indicate a user input step during processing, either entering and checking values or clicking.*

**Figures 4 and 5: difficult to read and not necessary. I suggest to remove or to rework in a more concise and readable way**

We feel that these figures are useful and provide the user with an idea of how running the algorithm actually looks. This goes into our point about high level of detail in the manuscript so the interested user can follow along very well upon close reading. In a digital version of the study, these images can be zoomed into, which yields high quality vector and raster graphics (tested at 300% zoom in Adobe Acrobat Reader).

**Concerning "5 Calibration and Validation Test I: Controlled Experiment": shorten and get to the point. The part concerning the cameras is not useful. What is important is the result (description of the Photoscan parameters is useless for example). Size of pixels do not**

**matter. What is important is the ratio between the resolution of the image to the size of the smallest grain detected.**

We disagree with the reviewer here, and feel that the discussion of camera types and our experimental setup is very useful to users that will want to apply the method directly and repeat the processing for their own field sites. This again goes towards the thoroughness of our study, where we have left none of our processing steps out.

**Same for "6 Calibration and Validation Test II: Field Surveys". I suggest to remove the useless details and to go to the point. You could show only the better and the worth examples and discuss why the "best" example give good results and why the "worst" example give no such good results (but good anyway ïA¿Ł )**

We feel that a close reading of the section demonstrates the good and bad results and the reasons for them. One example we point to here on P19, L7-9:

*Importantly, S24 is the only site not from a major river stem, but rather from a debris-flow fan draining a small tributary catchment in the Quebrada del Toro. S34 also had a high Adiff=−2.11. In this case, poor performance is due to significant blurriness of this image, and again a small sample size (n=47).*

**Figure 19 (they are too many figures): for me, what is important is to discuss why its work or not, in what case and How I can use your software and what error can I expect, by adding some advices on the acquisition procedure I should follow. These points are discussed in the current version but are not enough highlighted.**

We feel that this figure is demonstrative of the difference in the AIF and KMS routines and very instructive to the end user concerned about image quality and how it will affect the results from each technique. Regarding advice for acquisition, we have rewritten a large part of the discussion in response to the first review (see above). The point cloud integration has been removed which should add to the discussion clarity, and we end the discussion with a clearly outlined section of the "Practical Considerations for Image Collection and Processing".

---

## Editor Decision (ED1)

Dear Benjamin Purinton,

Thank you for submitting a new version of your manuscript entitled "Introducing PebbleCounts: A grain-sizing tool for photo surveys of dynamic gravel-bed rivers" to ESurf.

I have now received a second report from Patrice Carbonneau. He feels that the manuscript is much improved. Yet he reports a detailed list of small issues which need to be resolved before the paper can be published in ESurf. In particular, I do agree with him that a table summarizing the discussion about the relative performance of your approach would help to synthesize your message.

I have noted your resistance to take into account several of the comments raised by Pascal Allemand. Yet, I agree with his observation that the manuscript is difficult to follow in many places. Esurf is dedicated to a broad audience of scientists working on Earth surface processes. In its present version, the manuscript is closer to a technical report aiming at a more specialized community. Several points contribute to this feeling:
- the manuscript contains too many figures (18);
- it makes use of too many acronyms (GSD, SfM-MVS, MP, AIF, KMS, NMAD, ….),
- the level of sectioning is too high (8.3.1, etc…) and many of the section titles are inappropriate ('previous work on photo sieving', 'Motivations for new methods', `7.4.3. Results: Handlhed Image', etc… ).

Here are more specific comments:
• Figure 4 and 5 are certainly appropriate for a software manual. But they are not suitable for a journal like Esurf. I strongly advise you to move them into the supplementary information.
• Again, many of your section titles are inappropriate for a scientific journal. Here are a few exemples: « 2. Previous work on photo sieving », « Motivation for new methods », « 4. Additional Data dimensions from point clouds », « 5. The algorithms »,…  In many cases, you can solve this problem by suppressing these sections and merging them into a larger one with a broader title. As an example, section 2 and (maybe) 3 could be included as part of the introduction. Similarly, there is no need to divide subsection 6.3 (the title of which « images » is again quite clumsy)  in 2 subsubsections. These 2 exemples illustrate a more generic problem that you must address.
• The accumulation of field sites on Figure 14 and 15 is to the detriment of the message. You should restrict these figures to a couple of emblematic results illustrating your message, and use the supplementary information file to present the integrality of your results.
• Although the caption of figure 1 starts with « Conceptual », this figure does not provide any conceptual information. It merely illustrates the difference of results obtained using different methods.

In conclusion, I understand the need for details, but details are sometimes to the detriment of clarity. I therefore encourage you to submit a suitably revised version of your manuscript taking into account the remaining issues. Upon submission, I will need to receive a response file that lists each of the comments and describes how the manuscript has been modified (or not) in response to those comments.

I look forward to receiving your revised manuscript.

Sincerely yours,

Eric Lajeunesse

---

## Author Response (AR2)

Reply to reviewers for manuscript (esurf-2019-20) submission to Earth Surface Dynamics:

Introducing PebbleCounts: A grain-sizing tool for photo surveys of dynamic gravel-bed rivers – Purinton and Bookhagen

Highlighted in **bold** are the reviewer comments followed by our point-by-point replies in regular text. Sentences that will be added or changed from the original manuscript are in *italics*. All changes will be made to the final manuscript submission following completion of the interactive review period.

Following both responses is a marked-up version of the newer manuscript compared with the previous. Since a lot of material was also moved to the supplement, the marked-up manuscript is also followed by the new marked-up supplement.

**Reviewer Patrice Carbonneau (Second Report)**

**There remains significant issues with the question of the angle of views. First, top-down is a colloquial term and I remain firm that is should not be used in literature. Nadir is the correct term. Form most of the history of aerial imagery, there has always been error in the angle of acquisition. When aircraft, including drones, try to collect nadir data, there will still be fluctuations around the target angle of view. Having errors in that view is not a good reason to use the wrong nomenclature. If the authors are concerned that the camera mast setup induces error, then their images could be described as 'slightly off-nadir' or 'near-vertical' but nadir would still be fine because it is understood that no airborne camera can be controlled to have perfect orientation. This actually raises a question: To what extent are these images off-nadir? A number is not provided. In the current revision, there is in fact a problem of clarity in this case. When using the camera mast (or in the lab) are the authors deliberately trying to acquire images that are significantly oblique, i.e. actively tilting the camera off-nadir? If not, this is a nadir setup and the authors need to discuss the off-nadir component as a measurement error. This could in fact play in their favour by mitigating the doming effect in SfM-photogrammetry (See the James and Robson in ESPL, 2014). The authors need to:**

**1- Stop using the term top-down**

**2- Clarify if they are actively taking oblique imagery.**

We understand the concern of the reviewer with our terminology and think we can provide a solution by using the terms near-nadir, off-nadir, and oblique. We have deliberately chosen the term top-down and would like to refrain from using the term nadir, because these images were not always taken in nadir direction. As geologists and geophysicists, we are aware of the clear and precise definition of nadir (and zenith). These are perpendicular to an equipotential surface (for example the geoid) and nadir refers to an observation method that usually points in the direction of the force of gravity. This geometry is achieved by satellites and airplanes and is

often verified by additional measurements taken at the same time (in the case of historical aerial photography, level measurements with water bubbles were taken at the same). While we would have liked to achieve this geometry and we were striving for it, our setup does not guarantee that all of our photos were taken in precise nadir position. They may have been off by a few degrees to any side.

To clarify: We always tried to have the camera pointed directly downwards with the mast setup, but over the course of several hundred to thousand photos collected by hand, there were inevitably minor tilts ($< 10°$). It is true that these slightly oblique photos likely contributed to improved quality of the point clouds, but it was not our intention to tilt the camera significantly, and instead relied on the ground control targets to limit doming-effects. In addition, for the handheld lab setup, we intentionally collected ~10 images at a ~20° off-nadir angle, to get the sides of the pebbles in addition to their tops. In light of these points, we have changed top-down to near-nadir, off-nadir, or oblique throughout the manuscript (and in the figures), and changed the explanatory text in our new "Section 3.2. Orthomosaic Generation" (formerly part of a subsection "6.3.1. Top-Down Images") P8, L6:

*For each test setup, we collected ~10 images from ~20° off-nadir (oblique) and at least 4 overhead near-nadir (tilts < 10°) pictures, for 12-16 photos in total. The collection of oblique images aided in removing doming effects from the resulting point clouds (e.g., James and Robson, 2014) and for capturing the pebble edges and sides (Fig. S1).*

In reference to the mast photos, we have clarified at P12, L6 with the sentence:

*We refer to the images as near-nadir, rather than nadir, due to the fact that during mast photo collection some unintentional tilting of the camera (< 10°) occurred. These near-nadir photos aided in removing doming effects, but did not allow us to capture the sides of pebbles as in the oblique images taken in the experimental setup (Fig. S1). Capturing oblique images of every patch in the field sites would require infeasible amounts of time and processing power.*

**The authors have much improved the discussion of their errors in relation to other published work. But their refusal to provide a summary table is not justified nor is it reasonable. The authors are right that different papers report error differently, but that is not a good reason not to summarise information. There is abundant precedent for such tables in other similar works (see Dugdale et al in RRA on 'Aerial Photosieving'). The point is to give a sense, even if qualitative, of relative performance. The textual discussion of this topic is very good. But, I repeat, please provided a summary table, many of your readers will appreciate the condensed information and better absorb your points in this manner.**

A table has been added to the first discussion section (Table 1). There we briefly summarize the results of other authors who used similar segmentation methods. We do not include comparisons with texture-based methods, given the significant differences between segmentation- and texture-based methods, and the many differences between each texture method (semivariance, roughness, wavelets, etc.). We also add an explanation in the discussion P20, L2:

*For comparison of our algorithms to previous work, we do not consider errors reported in studies using texture-based measurements (e.g., Woodget et al., 2018), since these are based on correlative relationships rather than physical measurement of each grain. Texture methods work well for homogeneous pebble arrangements in lower-energy settings, but high-energy mountain rivers with heterogeneous pebble arrangements and large ranges in sizes require segmentation approaches.*

**The discussion of UAV applications is also much improved. But section 8.3.2 should be just 1 paragraph. The second paragraph repeats most of the ideas of the first and it seems like the first was dropped in without a proper reading of the second. This just needs to be merged and have repetitions smoothed out.**

We appreciate the close re-reading of our manuscript and have changed this section to read P22, L25:

*The > 20 m flight heights typical of UAV surveys lead to cm-scale imagery with currently available 12–24 megapixel cameras, which is less appropriate for PebbleCounts processing, unless large (> 0.2 m) cobbles and boulders dominate the river site. Acquiring 0.5 mm/pixel imagery from a DJI Mavic drone with a 12 megapixel camera requires a very low flight height of ~1.4 m, giving a field of view of only ~1.5x2 m. This may be improved using better cameras like on the Mavic 2 Pro (20 megapixel camera), but gathering such imagery with the high overlap (~80%) required for SfM-MVS processing is still difficult, particularly given current ~20-minute flight length limitations from available batteries. Given continual technology improvements (e.g., greater battery life, more accurate geo-tags from onboard dGPS, higher megapixel cameras, and reduced motion blur), it is within reason to expect hectare to multi-hectare SfM-MVS UAV surveys at mm resolution in seamless orthomosaics along entire river reaches in the near future. But, for the time-being, a single, non-overlapping orthoimage workflow proposed by Carbonneau et al. (2018) has high potential to achieve large-areal results. Their workflow, building on Carbonneau and Dietrich (2017), uses a number of high and oblique overlapping flights to orthorectify a lower non-overlapping flight with mm-scale acquisition, with resulting single, scaled images passed to Basegrain, or, alternatively, to PebbleCounts.*

Associate Editor Eric Lajeunesse

**Thank you for submitting a new version of your manuscript entitled "Introducing PebbleCounts: A grain-sizing tool for photo surveys of dynamic gravel-bed rivers" to ESurf.**

**I have now received a second report from Patrice Carbonneau. He feels that the manuscript is much improved. Yet he reports a detailed list of small issues which need to be resolved before the paper can be published in ESurf. In particular, I do agree with him that a table summarizing the discussion about the relative performance of your approach would help to synthesize your message.**

We have noted this concern and have inserted Table 1 in the discussion section, as mentioned in our above response to Patrice Carbonneau. This summarizes the results of our study in comparison with other similar segmentation-based pebble counting techniques.

**I have noted your resistance to take into account several of the comments raised by Pascal Allemand. Yet, I agree with his observation that the manuscript is difficult to follow in many places. Esurf is dedicated to a broad audience of scientists working on Earth surface processes. In its present version, the manuscript is closer to a technical report aiming at a more specialized community. Several points contribute to this feeling:**

To preserve the detailed information, some of the original manuscript has now been moved to the supplementary information. For instance, we have placed the detailed Agisoft processing steps in a new supplement Section S4. See the marked-up revised manuscript and supplement at the end of this document for further details. We agree with and support publishing a widely accessible manuscript, but feel that providing details of processing steps are important for generating reproducible science.

**- the manuscript contains too many figures (18);**

We appreciate this point and have made an effort to reduce the number of figures by moving four to the supplement. There are now 14 figures in the manuscript.

**- it makes use of too many acronyms (GSD, SfM-MVS, MP, AIF, KMS, NMAD, ….),**

We have changed GSD to grain-size distribution and MP to megapixel throughout the manuscript. We think that the acronyms UAV (unmanned aerial vehicle), SfM-MVS (structure-from-motion with multi-view stereo) and NMAD (normalized median absolute difference) are commonly used within the quantitative geomorphic community, and we feel the broad audience should have exposure to them. Our use of KMS (k-means with manual selection) and AIF (automatic with image filtering) short names to distinguish our two techniques is necessary and widely used through the entire manuscript and supplement. These acronyms make the text more

readable, but we have tried to carefully go through and remove any unnecessary or redundant usage of acronyms.

**- the level of sectioning is too high (8.3.1, etc…) and many of the section titles are inappropriate ('previous work on photo sieving', 'Motivations for new methods', `7.4.3. Results: Handlhed Image', etc… ).**

To reduce the sections, we have combined many subsections and also given the resulting sections more broad and consistent names.

**Here are more specific comments:**

**• Figure 4 and 5 are certainly appropriate for a software manual. But they are not suitable for a journal like Esurf. I strongly advise you to move them into the supplementary information.**

These figures are now in the supplementary material.

**• Again, many of your section titles are inappropriate for a scientific journal. Here are a few exemples: « 2. Previous work on photo sieving », « Motivation for new methods », « 4. Additional Data dimensions from point clouds », « 5. The algorithms »,… In many cases, you can solve this problem by suppressing these sections and merging them into a larger one with a broader title. As an example, section 2 and (maybe) 3 could be included as part of the introduction. Similarly, there is no need to divide subsection 6.3 (the title of which « images » is again quite clumsy) in 2 subsubsections. These 2 exemples illustrate a more generic problem that you must address.**

We have merged a number of subsections into larger sections and also moved some sections (e.g., Point-Cloud Dimensions) to the supplement. See the marked-up revised manuscript and supplement at the end of this document for further details.

**• The accumulation of field sites on Figure 14 and 15 is to the detriment of the message. You should restrict these figures to a couple of emblematic results illustrating your message, and use the supplementary information file to present the integrality of your results.**

We have moved Figure 14 with the site curves separated to the supplement (Figure S7). Figure 11 shows the aggregated results and the reader is now referred to the new Section S5 in the supplement for the separate results. We prefer to keep all of the data points in Figure 15 (now Figure 12), since this shows the final errors associated with the methods (and referenced in our new Table 1). Although this figure is dense, it is important for understanding the error

distribution between both methods. We refer the reader to the new Section S5 and Figure S7 in the supplement for the detailed curves from each site showing where these data points originated.

**• Although the caption of figure 1 starts with « Conceptual », this figure does not provide any conceptual information. It merely illustrates the difference of results obtained using different methods.**

We have removed the term conceptual in reference to this figure.

**In conclusion, I understand the need for details, but details are sometimes to the detriment of clarity. I therefore encourage you to submit a suitably revised version of your manuscript taking into account the remaining issues. Upon submission, I will need to receive a response file that lists each of the comments and describes how the manuscript has been modified (or not) in response to those comments.**

We appreciate the reviewers' and your effort to improve the manuscript quality. These comments are well taken and we have shortened our manuscript accordingly. We would like to point out that the motivation for our detailed descriptions were to generate a reproducible scientific product that can applied elsewhere. We see now that some of our efforts provided too much detail and have subsequently moved material to the supplementary material for the interested reader. See the following marked-up manuscript and supplement for details on our revisions. We hope that the manuscript is now more readable to a general audience while also retaining the vital details to follow along step-by-step.

Sincerely,

From both authors,

Benjamin Purinton

[revised manuscript text omitted]
 for the indoor handheld imagery (with field-gathered mast imagery differences in parantheses following the step):

1. Image quality detection and the exclusion of photos with quality metric < 0.7. This step analyzes pixel contrast to estimate sharpness with values ranging from 0/blurred to 1/sharp. We found 0.7 to be a sufficient lower cutoff upon visual inspection of results.

2. Detection of 12-bit coded targets in the remaining photos, with two targets placed at each of the four corners of the area and ensuring that the diameter of the printed targets' center circle was limited to 10–30 pixels in image resolution for successful automated detection.

3. Input of scale for the orthomosaic output, provided by the distances between the targets at each corner, resulting in four distance measurements, with 0.5 mm accuracy using a ruler with cm and mm demarcations. (For the field images: The scale was provided by the XYZ coded target locations in UTM zone 19S, WGS84 ellipsoidal datum.)

4. Photo alignment at high quality with a 40,000 key-point and 2000 tie-point limit.

5. Dense cloud generation from the aligned photos at the medium output and with moderate depth filtering. Given the high quality of the photos more aggressive options did not improve results. (For the field images: Given the increased complexity of the setting and imperfect photo collection, the dense point cloud was generated at high quality with aggressive depth filtering.)

6. DEM building from the dense cloud with default settings in a local coordinate system. (For the field images: The DEMs and orthomosaics were also output in UTM zone 19S projections, providing undistorted pixels with resolution in m/pixel.)

7. Generation of an orthomosaic using the DEM for orthorectification at the default settings.

8. Output of the orthomosaic to a GeoTiff file with resolution provided in m/pixel.

**S5. KMS and AIF Results Separated by Site**

Here we show all of the results (following 20-pixel truncation) for each of the 12 sites in Figure S7. These results are aggregated in curves shown in the main manuscript Figure 11 and a comparison of the individual percentiles of interest is shown in the main manuscript Figure 12.

[Figure]

**Figure S7.** Comparison of 20-pixel truncated grain-size distributions between hand-clicked control (black line), KMS *PebbleCounts* (gray, dashed line), and AIF *PebbleCountsAuto* (red, dotted line) for the $12 \times {\sim}1.16$ mm/pixel control sites. In corresponding colors are the $p$-value results of a KS-test and the $A_{diff}$ approximate integral between the curves for each approach versus the control data. The legend indicates the number of grains ($n$) making up each curve. See Figure 6b in the main manuscript for sites.

**S6. Misidentification in the AIF Approach**

Figure S8 demonstrates remaining issues with the AIF approach in a few map-view examples. On a grain-by-grain basis, there are many inaccuracies falling into three main categories: over-segmentation of grains with internal edges and the selection of each segment as a separate grain, under-segmentation and merging of neighboring grains that have weak edges sometimes caused by image blur, and misidentification of non-grain objects or clusters of small grains. It is clear from this analysis that caution must be used when interpreting AIF results, particularly in complex or blurry images.

[Figure]

**Figure S8.** Resulting delineated grains using the AIF *PebbleCountsAuto* function (top row) versus the same area from the KMS *PebbleCounts* function (bottom row). Labels indicate the issues with the AIF results and improvement in KMS results. Note the poor results for the blurry image on the right (S34).